# LUMOS$\mathcal{X}$: RELATE ANY IDENTITIES WITH THEIR ATTRIBUTES FOR PERSONALIZED VIDEO GENERATION

**Jiazheng Xing**[*,1,4,2], **Fei Du**[*2,3], **Hangjie Yuan**[*2,3,1], **Pengwei Liu**[1,2], **Hongbin Xu**[4],
**Hai Ci**[4], **Ruigang Niu**[2,3], **Weihua Chen**[†2,3], **Fan Wang**[2], **Yong Liu**[†1]

[1]Zhejiang University, [2]DAMO Academy, Alibaba Group, [3]Hupan Lab, [4]National University of Singapore

* Equal contribution, [†] Corresponding authors.

`jiazhengxing@zju.edu.cn, yongliu@iipc.zju.edu.cn`

Project Page: `https://jiazheng-xing.github.io/lumosx-home/`

## ABSTRACT

Recent advances in diffusion models have significantly improved text-to-video generation, enabling personalized content creation with fine-grained control over both foreground and background elements. However, precise face–attribute alignment across subjects remains challenging, as existing methods lack explicit mechanisms to ensure intra-group consistency. Addressing this gap requires both explicit modeling strategies and face-attribute-aware data resources. We therefore propose *Lumos$\mathcal{X}$*, a framework that advances both data and model design. On the data side, a tailored collection pipeline orchestrates captions and visual cues from independent videos, while multimodal large language models (MLLMs) infer and assign subject-specific dependencies. These extracted relational priors impose a finer-grained structure that amplifies the expressive control of personalized video generation and enables the construction of a comprehensive benchmark. On the modeling side, Relational Self-Attention and Relational Cross-Attention intertwine position-aware embeddings with refined attention dynamics to inscribe explicit subject–attribute dependencies, enforcing disciplined intra-group cohesion and amplifying the separation between distinct subject clusters. Comprehensive evaluations on our benchmark demonstrate that *Lumos$\mathcal{X}$* achieves state-of-the-art performance in fine-grained, identity-consistent, and semantically aligned personalized multi-subject video generation.

## 1 INTRODUCTION

In recent years, diffusion models (Ho et al., 2020; Ho & Salimans, 2022; Rombach et al., 2022) have driven remarkable progress, establishing new performance standards in text-to-video generation (Hong et al., 2022; Blattmann et al., 2023; Menapace et al., 2024; Zheng et al., 2024), particularly through the adoption of Diffusion Transformer (DiT) architectures (Peebles & Xie, 2023). These advances have laid a solid groundwork for customized video generation (Ma et al., 2024; Yuan et al., 2024; Zhong et al., 2025; Chen et al., 2025; Huang et al., 2025; Fei et al., 2025; Liu et al., 2025), where high-degree-of-freedom personalization unlocks transformative applications ranging from virtual theatrical production to e-commerce—enabling fine-grained control over both backgrounds and foregrounds, including multiple interacting subjects. Yet, realizing open-set personalized multi-subject video generation under such flexible and complex conditions remains profoundly challenging. The task requires not only the precise integration of diverse and interrelated conditioning signals but also the preservation of temporal coherence and identity fidelity across all subjects.

In the realm of open-set personalized video generation, prior studies have pushed the field forward from distinct angles. Certain approaches (Ma et al., 2024; He et al., 2024; Yuan et al., 2024; Zhang et al., 2025b; Zhong et al., 2025) concentrate narrowly on foreground facial customization, preserving identity fidelity from reference images yet affording only limited flexibility in input specification. In contrast, more recent methods (Chen et al., 2025; Huang et al., 2025; Fei et al., 2025; Liu et al., 2025) enable highly versatile multi-subject video personalization with controllable foregrounds and backgrounds, but they largely neglect the intrinsic dependency structures that govern multi-subject

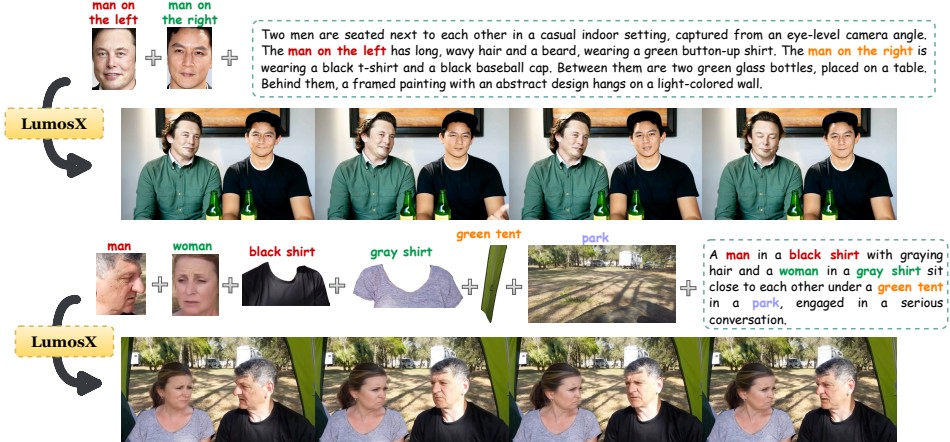

Figure 1: Lumos$\mathcal{X}$ supports flexible personalized multi-subject video generation.

conditions. Crucially, during fine-grained multi-condition injection, conditioning signals for each subject are typically decomposed into facial exemplars and attribute descriptors (*e.g.*, *man: blond hair, white T-shirt, sunglasses*). Absent an explicit mechanism to bind identity with its associated attributes, such formulations are inherently fragile and frequently yield attribute entanglement or face–attribute misalignment across subjects.

Although implicit modeling via textual captions can capture simple multi-subject dependencies during video generation, ambiguity often arises when captions contain similar subject nouns, such as "*A man on the left with ... and a man on the right with ...,*" leading to confusion in subject–attribute associations. To overcome this limitation, under fine-grained multi-subject inputs, explicit constraints must be imposed at both the **data** and **model** levels. (1) **Data level**: When visual references are provided, the correspondence between each face and its associated attributes should be clearly specified. (2) **Model level:** During generation, each face-attribute pair is explicitly bound into an independent subject group, with intra-group correlation enhanced and inter-group interference suppressed.

To address the challenge of modeling face-attribute dependencies in multi-subject video generation, we present ***Lumos$\mathcal{X}$***, a novel framework for personalized multi-subject synthesis. On the data side, the absence of public datasets with annotated dependency structures motivates us to construct a collection pipeline that supports open-set entities. This pipeline extracts captions and foreground–background visual conditions from independent videos, while multimodal large language models (MLLMs) infer and assign subject-specific dependencies. In particular, it produces customized single- and multi-subject data with explicit face–attribute correspondences, which not only enhance personalization during modeling but also enable the construction of a comprehensive benchmark. On this basis, the benchmark further defines two evaluation tasks, **identity-consistent** and **subject-consistent** generation, which allow a systematic assessment of a model's ability to preserve identity and align multi-subject relationships. On the modeling side, *Lumos$\mathcal{X}$* explicitly encodes face-attribute bindings into coherent subject groups through two dedicated modules: **Relational Self-Attention** and **Relational Cross-Attention**. The Relational Self-Attention module incorporates Relational Rotary Position Embedding (R2PE) and a Causal Self-Attention Mask (CSAM) to model dependencies at the positional encoding and spatio-temporal self-attention stages. In addition, the Relational Cross-Attention module introduces a Multilevel Cross-Attention Mask (MCAM), which reinforces intra-group coherence, suppresses cross-group interference, and refines the semantic representation of visual condition tokens. *Lumos$\mathcal{X}$* is built upon the *Wan2.1* (Wang et al., 2025) text-to-video backbone, with our modules seamlessly integrated to support flexible and high-fidelity personalized multi-subject generation, as shown in Fig. 1. Extensive experiments demonstrate *Lumos-$\mathcal{X}$*'s strong capability in producing fine-grained, identity-consistent, and semantically aligned personalized videos, achieving state-of-the-art results across diverse benchmarks.

The contributions of our *Lumos$\mathcal{X}$* can be summarized as follows:

- **Data Side.** We build a collection pipeline for open-set multi-subject generation that extracts captions and foreground–background condition images with explicit face–attribute depen-

dencies from independent videos. This yields finer-grained relational priors that enhance personalized video customization and enable the construction of reliable benchmarks.

- **Model Side.** We introduce Relational Self-Attention and Relational Cross-Attention, which integrate relational positional encodings with structured attention masks to explicitly encode face–attribute bindings. This reinforces intra-group coherence, mitigates cross-group interference, and ensures semantically consistent multi-subject video generation.

- **Overall Performance.** Through extensive experiments and comparative evaluations, *LumosX* achieves state-of-the-art results in generating fine-grained, identity-consistent, and semantically aligned personalized multi-subject videos, decisively outperforming advanced open-source approaches including Phantom and SkyReels-A2.

## 2 RELATED WORKS

**Video Generation.** Video generation has advanced rapidly in recent years, becoming one of the most dynamic research areas. Early works based on generative adversarial networks (GANs) (Vondrick et al., 2016; Tulyakov et al., 2018) demonstrated initial video synthesis but struggled with temporal coherence and fidelity. Latent diffusion models (LDMs) (Rombach et al., 2022), powered by UNet (Ronneberger et al., 2015), marked a significant milestone by enabling high-quality video generation through denoising in compressed latent spaces. These works typically add a temporal module to an image generation model, such as Make-A-Video (Singer et al., 2022) and Animatediff (Guo et al., 2023). However, these models often face scalability challenges when scaling to larger parameter sizes or higher resolutions. Diffusion Transformers (DiTs) (Peebles & Xie, 2023), replacing the Unet backbone with Transformer blocks, have shown superior performance in visual generation. By incorporating spatio-temporal attention mechanisms, video DiTs achieved unprecedented performance in modeling long-range dependencies across both spatial and temporal dimensions, significantly enhancing video realism and consistency. Models like Hunyuan Video (Kong et al., 2024), Wan2.1 (Wang et al., 2025), and MAGI-1 (Sand-AI, 2025) have scaled the parameters of video DiTs to more than 10 billion, achieving significant advancement. Despite these advances, controllability remains a critical bottleneck: text-driven generation often fails to precisely align with user intentions due to ambiguities in natural language descriptions. This work focuses on multi-subject video customization to address this limitation, enabling precise content-control video generation.

**Multi-Subject Video Customization**. In recent years, subject-driven video generation has attracted growing interest. Several works focus on ID-consistent video generation, such as Magic-Me (Ma et al., 2024), ID-Animator (He et al., 2024), ConsisID (Yuan et al., 2024), Magic Mirror (Zhang et al., 2025a), FantasyID (Zhang et al., 2025b), and Concat-ID (Zhong et al., 2025). These works generate videos that show consistent identity with the reference images, mainly focusing on facial identity. For arbitrary subject customization, VideoBooth (Jiang et al., 2024) incorporates high-level and fine-level visual cues from an image prompt to the video generation model via cross attention and cross-frame attention. DreamVideo (Wei et al., 2024) customizes both subject and motion, with motion extracted from a reference video. Although they have demonstrated capabilities in generating single-subject consistent videos, neither the data processing nor the model can be easily transferred to the more challenging multi-subject customization. CustomVideo (Wang et al., 2024c) generates multi-subject identity-preserving videos by composing multiple subjects in a single image and designs an attention control strategy to disentangle them. However, it requires test-time finetuning for different subjects. Recently, several works (Chen et al., 2025; Huang et al., 2025; Fei et al., 2025; Liu et al., 2025) propose to customize multiple subjects in the video DiTs. Different subjects are usually concatenated and fed into the video DiT network without distinguishing between them. This lack of differentiation can lead to semantic ambiguity, especially when there are numerous targets and hierarchical relationships among them. In this work, we design several strategies to differentiate between various subjects and their hierarchical relationships, achieving consistent customization while ensuring harmony among different objectives and the text's adherence capabilities.

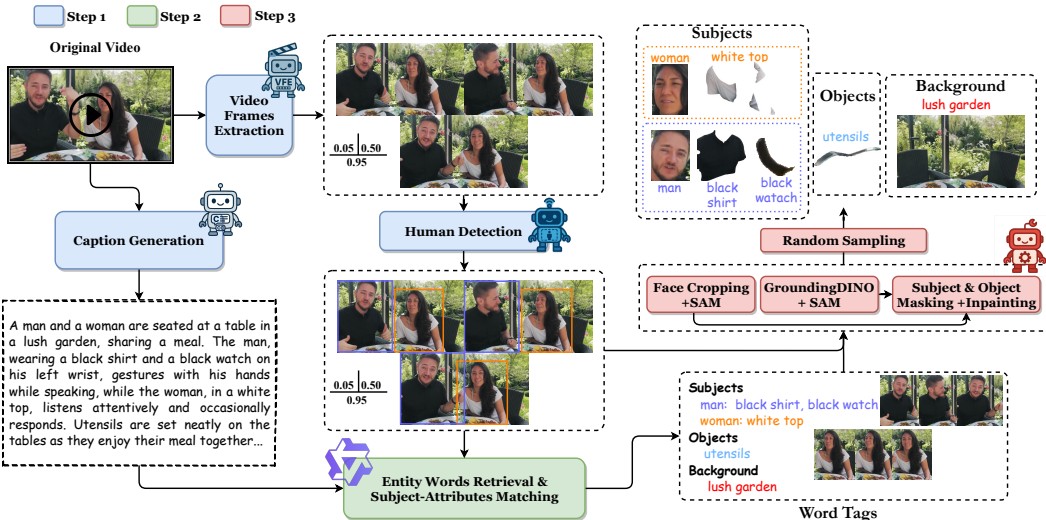

Figure 2: **Dataset construction pipeline for personalized multi-subject video generation.** We build the training dataset from raw videos in three steps: (1) generate a caption and detect human subjects in extracted frames; (2) retrieve entity words from the caption and match subjects with their attributes; (3) use these entity tags to localize and segment target subjects and objects, producing a clean background image.

## 3  METHODS

### 3.1  PRELIMINARY

In this work, we bulid upon the latest text-to-video generative model, *Wan2.1* (Wang et al., 2025), which comprises the 3D variational autoencoder (VAE) $\mathcal{E}$, the text encoder $\mathcal{T}$, and the denoising DiT (Peebles & Xie, 2023) backbone $\epsilon_\theta$ combined with Flow Matching (Lipman et al., 2022). Within the DiT architecture, full spatio-temporal Self-Attention is used to capture complex dynamics, while Cross-Attention is employed to incorporate text conditions. Specifically, given a video $X = \{x_i\}_{i=1}^{N}$ with $N$ frames, $\mathcal{E}$ compresses it into a latent representation $\mathbf{z} \in \mathbb{R}^{T \times HW \times C}$ along the spatiotemporal dimensions, where $T$, $HW$, and $C$ denote the temporal, spatial, and channel dimensions, respectively. The text encoder $\mathcal{T}$ takes the text prompt and encodes it into a textual embedding $\mathbf{c}_{\text{text}}$. The denoising DiT $\epsilon_\theta$ then processes the latent representation $\mathbf{z}$ and the textual representation $\mathbf{c}_{\text{text}}$ to predict the distribution of video content. Each DiT Block incorporates 3D Rotary Position Embedding (3D-RoPE (Su et al., 2024)) within the full spatio-temporal attention module to better capture both temporal and spatial dependencies.

### 3.2  DATASET CONSTRUCTION

As illustrated in Fig. 2, our training dataset and inference benchmark for personalized multi-subject video generation are constructed from raw videos through the following three steps.

**Caption Generation and Human Detection.**  To obtain richer textual descriptions for downstream tasks, we replace the original video captions with captions generated by the large vision–language model VILA (Lin et al., 2024). We sample three frames from the beginning, middle, and end of each video (5%, 50%, and 95% positions) and apply human detection (Wang et al., 2024a) to extract human subjects for subsequent face–attribute matching.

**Entity Words Retrieval and Face-Attribute Matching.**  In this step, our goal is to retrieve entity words from the caption, which can be classified into three categories: human subjects with attributes (*e.g.*, *man: black shirt, black watch*), objects (*e.g.*, *utensils*), and background (*e.g.*, *lush garden*). During this process, if multiple human subjects are present, we need to assign different attributes to the corresponding subjects. In particular, when the caption contains multiple instances of the same subject noun (*e.g.*, *woman*), we rely on visual information to assist in distinguishing between them. Therefore, we employ the multimodal large language model Qwen2.5-VL (Bai et al., 2025)

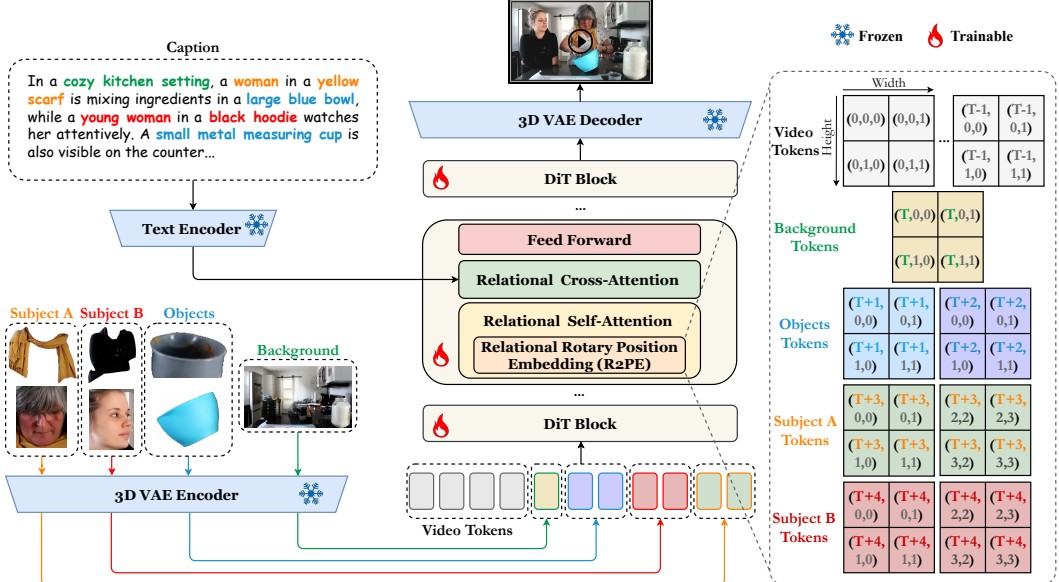

Figure 3: **Overview of Lumos𝒳.** Built on the T2V model *Wan2.1* (Wang et al., 2025), our framework encodes all condition images into image tokens via a VAE encoder, concatenates them with denoising video tokens, and feeds the result into DiT (Peebles & Xie, 2023) blocks. Within each block, the proposed Relational Self-Attention and Relational Cross-Attention enable causal conditional modeling, enhance visual token representations, and ensure precise face–attribute alignment.

to retrieve multiple entity words from the caption, while leveraging prior visual information from human detection results to achieve precise face-attribute matching.

**Obtaining Condition Images.** For subjects, we apply face detection (Wang et al., 2024a) within human detection boxes to extract face crops and use SAM (Kirillov et al., 2023) to segment attribute masks. For objects, GroundingDINO (Liu et al., 2024) combined with SAM segments each entity within the global image. For backgrounds, we remove subjects and objects using the crops and masks, then apply the diffusion inpainting model FLUX (Labs, 2023) to generate a clean background. Finally, from the valid results of the three key frames, we randomly select one per entity as its condition image—matching the inference process, where each condition uses a single reference image—while ensuring data diversity by preventing all selections from a single frame.

Through these three steps, we obtain the visual condition images for the subjects, objects, and background, along with their paired word tags derived from the input text caption. Note that a subject is defined as a single human face paired with its corresponding attributes. The face is expected to present clear facial features without significant occlusion, and the associated attributes can include clothing (top or bottom), accessories (*e.g.*, glasses, earrings, or necklaces), or hairstyle.

## 3.3 Lumos𝒳

As shown in Fig. 3, our framework builds on the T2V model *Wan2.1* (Wang et al., 2025). To enable personalized multi-subject video generation, all condition images are encoded into image tokens via a VAE encoder, concatenated with denoising video tokens, and fed into DiT (Peebles & Xie, 2023) blocks. Within each block, we introduce **Relational Self-Attention** with Relational Rotary Position Embedding (R2PE) and a Causal Self-Attention Mask to support spatio-temporal and causal conditional modeling. Additionally, **Relational Cross-Attention** with a Multilevel Cross-Attention Mask (MCAM) incorporates textual conditions, strengthens visual token representations, and aligns face–attribute relationships.

### 3.3.1 RELATIONAL SELF-ATTENTION

**Relational Rotary Position Embedding (R2PE).** In T2V models like *Wan2.1* (Wang et al., 2025), utilizing 3D Rotary Position Embedding (3D-RoPE) to assign position indices $(i, j, k)$ to video

tokens is necessary, which can affect the interaction among these tokens. In T2V tasks, the original 3D-RoPE assigns position indices $(i, j, k)$ sequentially to the video tokens $\mathbf{z} \in \mathbb{R}^{T \times HW \times C}$, where $i \in [0, T)$, $j \in [0, W)$, and $k \in [0, H)$. In personalized multi-subject video generation tasks, it is essential to not only extend 3D-RoPE to the reference condition images but also to preserve the face-attribute dependency throughout this process.

Given the concatenated VAE tokens $\mathbf{z}' = [\mathbf{z}; \mathbf{z}_c] \in \mathbb{R}^{(T+N_c) \times HW \times C}$, where $\mathbf{z_c} \in \mathbb{R}^{N_c \times HW \times C}$ represents the condition tokens, we introduce the Relational Rotary Position Embedding (R2PE), as illustrated in Fig. 3. The condition tokens $\mathbf{z}_c$ are composed of subject tokens $\mathbf{z}_{sub}$, object tokens $\mathbf{z}_{obj}$, and background tokens $\mathbf{z}_{bg}$, *i.e.*, $\mathbf{z_c} = [\mathbf{z}_{sub}; \mathbf{z}_{obj}; \mathbf{z}_{bg}]$. In R2PE, for the video tokens $\mathbf{z}$, we adopt the standard 3D-RoPE $(i, j, k)$ position assignment method, while for the background $\mathbf{z}_{bg}$ and object $\mathbf{z}_{obj}$ tokens, we sequentially extend each entity along the $i$-index. For the subject tokens $\mathbf{z}_{sub}$, which are composed of human face tokens $\mathbf{z}_{face}$ and human attribute tokens $\mathbf{z}_{attr}$, we strictly adhere to the face-attribute dependency when assigning position indices to the subject tokens. Therefore, for the human face tokens and their corresponding attribute tokens within the same group, they share the same $i$-index and are extended along the $j$-index and $k$-index. Specifically, the position index for the condition tokens $\mathbf{z}_c$ is defined as:

$$(i', j', k') = \begin{cases} \left( i_{bg/obj} + T, j, k \right), & when \ \mathbf{z}_{bg} \ and \ \mathbf{z}_{obj} \\ \left( i_{sub} + T + N_{bg/obj}, j + W * N^g_{i_{sub}}, k + H * N^g_{i_{sub}} \right), & when \ \mathbf{z}_{sub} \end{cases}$$

(1)

where $i_{bg/obj} \in [0, N_{bg/obj})$, with $N_{bg/obj}$ denoting the total number of background and object entity. $i_{sub} \in [0, N_{sub})$ where $N_{sub}$ represents the total number of face-attribute subject groups. And, $N^g_{i_{sub}} \in [0, N_{i_{sub}})$, where $N_{i_{sub}}$ denotes the total number of face and attribute entity within the $i_{sub}$ subject group. The proposed R2PE effectively inherits and extends the implicit positional correspondence of the original *Wan2.1* model, while preserving the face-attribute dependency within each group of the subject condition.

**Causal Self-Attention Mask (CSAM).** The Causal Self-Attention Mask is a boolean matrix, as illustrated in Fig. 4 (a), with the following two rules governing its mechanism: (I) Calculations are performed within each conditional branch, where the human face and its corresponding attributes are treated as a unified subject condition branch; (II) Video denoising tokens apply unidirectional attention to the condition tokens only. Given the concatenated tokens $\mathbf{z}' \in \mathbb{R}^{(T+N_c) \times HW \times C}$, the mask can be formulated as:

$$\mathbf{M}^{SA}_{\mathbf{q}, \mathbf{k}} = \begin{cases} \text{True}, & \text{if} \ \mathbf{q} \in \mathbf{z} \ \text{or} \ \mathbf{q} == \mathbf{k} \ \text{or} \ \mathbf{q}, \mathbf{k} \in \mathbf{z}^g_{sub} \\ \text{False}, & \text{otherwise} \end{cases}$$

(2)

where $\mathbf{q}$ and $\mathbf{k}$ denote the categories of the tokens corresponding to the query-key matrix in Self-Attention, both of which belong to the visual concatenated tokens $\mathbf{z}'$. $\mathbf{z}$ and $\mathbf{z}^g_{sub}$ represent the denoising video tokens and the face/attribute tokens within the same subject group. This causal mask enforces constraints on the range of interactions during the Self-Attention process. This design efficiently prevents unidirectional attention from the conditional branch to the denoising branch, while enabling the denoising branch to independently aggregate conditional signals and efficiently bind the face-attribute dependencies within the conditional branch. To enable efficient computation, we employ the MagiAttention mechanism proposed in (Sand-AI, 2025).

### 3.3.2 RELATIONAL CROSS-ATTENTION

**Multilevel Cross-Attention Mask (MCAM).** In the Cross-Attention process of the T2V task, all visual tokens interact with all textual tokens. However, the requirements may differ for customized video generation tasks. Intuitively, all textual tokens are of equal importance for video denoising tokens. However, for visual condition tokens in customized tasks, each has a corresponding textual token, such as: face image → "*man*". Therefore, we aim to enhance the interaction of visual condition tokens with the corresponding textual tokens in the cross-attention process to improve the semantic representation of visual tokens. Furthermore, for subject condition tokens, we seek to strengthen the face-attribute dependency within the same subject group in the Cross-Attention process, while reducing the mutual influence between different subject groups.

Based on the aforementioned motivation, we propose the Multilevel Cross-Attention Mask (MCAM), as shown in Fig. 4(b). MCAM is a numerical mask in which we have defined three

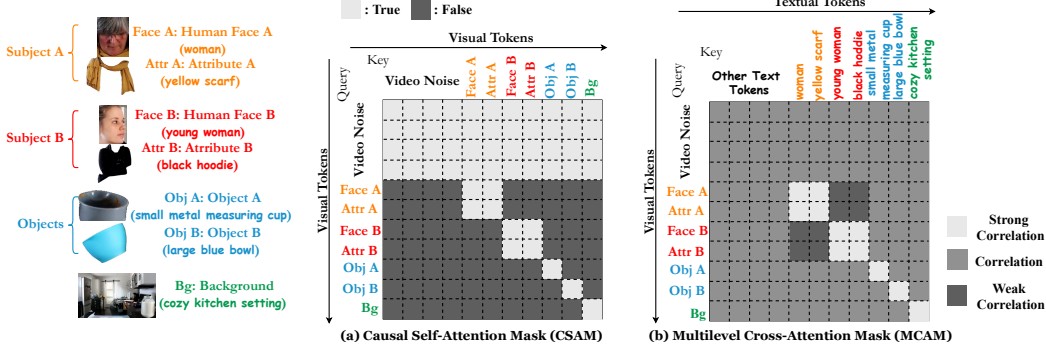

Figure 4: **Illustration of Attention Mask Design:** Causal Self-Attention Mask (CSAM) and Multilevel Cross-Attention Mask (MCAM). We present a specific customized task case as an example.

levels of correlation: Strong Correlation (1), Correlation (0), and Weak Correlation (-1). Specifically, Strong Correlation applies to the interaction between the visual condition tokens and their corresponding textual tokens, as well as between visual subject (face & attribute) condition tokens and all textual tokens within the same subject group. Weak Correlation applies to the interaction between visual subject tokens and the textual tokens from different subject groups. And all other cases remain as Correlation. Therefore, this mask $\mathbf{M}_{\mathbf{q},\mathbf{k}}^{CA}$ can be formulated as:

$$
\mathbf{M}_{\mathbf{q},\mathbf{k}}^{CA} = \begin{cases} 1 \,(\text{Strong Correlation}), & \text{if } \mathbf{q}, \mathbf{k} \text{ belong to the same semantic entity or subject group} \\ -1 \,(\text{Weak Correlation}), & \text{if } \mathbf{q}, \mathbf{k} \text{ belong to the different subject group} \\ 0 \,(\text{Correlation}), & \text{otherwise} \end{cases}
$$
(3)

where $\mathbf{q}$ and $\mathbf{k}$ denote the categories of the tokens corresponding to the query-key matrix in Cross-Attention, with the query and key representing the visual and textual tokens, respectively, in this context. Subsequently, we inject this constraint mask $\mathbf{M}_{\mathbf{q},\mathbf{k}}^{CA}$ into the Cross-Attention as follows:

$$
\text{Cross-Attention}(\mathbf{Q}, \mathbf{K}, \mathbf{V}) = \text{Softmax}\left(\frac{\mathbf{Q}\mathbf{K}^{\top} + \mathbf{M}_{\mathbf{q},\mathbf{k}}^{CA} \cdot \mathbf{s} \cdot r}{\sqrt{d_{\mathbf{K}}}}\right)\mathbf{V}
$$
(4)

where $\mathbf{Q}$ denotes concatenated visual features, and $\mathbf{K}$ and $\mathbf{V}$ are textual features. The hyperparameter $r$ controls the strength of the $\mathbf{M}_{\mathbf{q},\mathbf{k}}^{CA}$ constraint. Because similarity scores between query and key tokens vary across positions, a uniform mask template cannot be applied directly. To address this, we introduce a dynamic scaling factor $\mathbf{s}$ to adjust $\mathbf{M}_{\mathbf{q},\mathbf{k}}^{CA}$ at each position. The most straightforward strategy is to use the absolute value of the similarity matrix itself as $\mathbf{s}$. However, existing accelerated Attention computation modules based on Pytorch do not support customized numerical masks like this, and recomputing the similarity scores between $\mathbf{Q}$ and $\mathbf{K}$ outside the Attention module would incur significant computational overhead. To balance accuracy and efficiency, we propose an approximate method to compute the similarity matrix and derive $\mathbf{s}$ outside the Attention module as:

$$
\mathbf{s} = \text{Repeat}\left(\left|\mathbf{Q}_{ds}\mathbf{K}^{\top}\right|, \text{shape}\left(\mathbf{Q}\mathbf{K}^{\top}\right)\right)
$$
(5)

where $\mathbf{Q}_{ds}$ denotes $\mathbf{Q}$ downsampled by a factor of $d \times d$ via local average pooling on its spatial dimensions. The $\text{Repeat}(\cdot, \text{shape}(\cdot))$ operation restores the downsampled similarity matrix to its original size. Overall, the proposed MACM effectively strengthens relational dependency consistency and enhances the semantic-level representation of visual condition tokens.

# 4 EXPERIMENTS

## 4.1 EXPERIMENTAL SETTINGS

**Datasets.** Our personalized multi-subject video generation training dataset is built on Panda70M (Chen et al., 2024). After the cleaning and processing steps described in Sec. 3.2, we obtain 1.57M samples: 1.31M single-subject, 0.23M two-subject, and 0.03M three-subject videos.

**Benchmark.** For the testing benchmark, 500 videos crawled from YouTube are processed using the method described in Sec. 3.2, including 220 single-subject, 230 two-subject, and 50 three-subject videos. To rigorously assess personalized multi-subject video generation, we establish two tasks in this benchmark: identity-consistent and subject-consistent video generation. For **identity-consistent** video generation, 1 to 3 facial reference images are provided, with the facial similarity between the reference images and generated videos assessed using FaceSim-Arc (ArcSim), based on ArcFace (Deng et al., 2019), and FaceSim-Cur (CurSim), based on CurricularFace (Huang et al., 2020), following (Zhong et al., 2025) and (Yuan et al., 2024). We also employ VideoCLIPXL (ViCLIP-T) (Wang et al., 2024b) to measure semantic similarity between generated videos and text prompts. For **subject-consistent** video generation, the model takes multiple reference images, including faces, attributes, objects, and the background, as input. To comprehensively evaluate this task, we consider two aspects: 1) *Evaluation of the entire video*, where we use ViCLIP-T (Wang et al., 2024b) and ViCLIP-V (Wang et al., 2024b) to assess semantic similarity between the generated videos and text prompts, as well as between the generated videos and the ground-truth videos, respectively. Additionally, to prevent the generated videos from exhibiting copy-paste artifacts, we assess their dynamic degree, following (Huang et al., 2024). 2) *Evaluation on subjects*, where we first use Florence-2 (Xiao et al., 2024) to detect the person subjects based on the text prompts, and then apply OWLv2 (Minderer et al., 2023) to detect the bounding boxes of the person's attributes and other objects. Once all subjects are located, we compute CLIP-T (Radford et al., 2021) between the cropped image regions and the corresponding text prompts, as well as DINO-I (Oquab et al., 2023) and CLIP-I (Radford et al., 2021) between these regions and the corresponding reference images. ArcSim (Deng et al., 2019) is also used here to assess the identity similarities. If a subject is not detected in the generated video, the score is set to zero.

**Implementation Details.** Lumos$\mathcal{X}$ is fine-tuned from *Wan2.1*'s T2V(1.3B) (Wang et al., 2025) model, built on the DiT (Peebles & Xie, 2023) architecture. Video generation is performed at 480p resolution, with each training clip containing 81 frames (5 seconds at 16 FPS). In MACM, the downsampling factor $d$ and hyperparameter $r$ are set to 8 and 0.5, respectively. During training, each subject's face is associated with up to three attributes. Therefore, we recommend providing no more than three attributes per subject group during inference to maintain consistency with the training setup. Lumos$\mathcal{X}$ is trained in two phases: 15k iterations on single-subject data, followed by 16k iterations on mixed multi-subject data. Training uses the Adam (Kingma, 2014) optimizer with a learning rate of 1e-5, EMA decay of 0.99, weight decay of 1e-4, gradient clipping at 1.0, a batch size of 64, and random text-conditioning dropout at 10%. Our full training process required approximately 883 GPU-days on H20 GPUs. During inference, we use 50 steps and set the CFG scale to 6.

## 4.2 MAIN RESULTS

In the following experiments, all baseline configurations are fixed when compared against our Lumos$\mathcal{X}$ (*Wan2.1-1.3B*), including ConsisID (*CogVideoX-5B*) (Yuan et al., 2024), Concat-ID (*Wan2.1-1.3B*) (Zhong et al., 2025), SkyReels-A2 (*Wan2.1-14B*) (Fei et al., 2025), and Phantom (*Wan2.1-1.3B*) (Liu et al., 2025). For the Identity-Consistent Video Generation setting, all methods use the same inputs: each subject's face image and a shared global text prompt. For the Subject-Consistent Video Generation setting, we also enforce strict input parity. All methods receive exactly the same inputs as Lumos$\mathcal{X}$, including each subject's face image, all associated attribute images, object reference images, the background image, and the shared global prompt.

**Identity-consistent video generation.** In this experiment, we use only face reference images as input to evaluate identity preservation with ArcSim and CurSim. We first compare our Lumos$\mathcal{X}$ with face-specific customization methods, ConsisID (Yuan et al., 2024) and Concat-ID (Zhong et al., 2025), on a single-face test set of 220 videos, as ConsisID supports only single-face customization and Concat-ID has only released weights for the single-face setting, with results presented in Tab. 1. Additionally, we compare Lumos$\mathcal{X}$ with SkyReels-A2 (Fei et al., 2025) and Phantom (Liu et al., 2025), two general multi-subject video customization methods, on the full test set of 500 videos, as shown in Tab. 2. Across both settings, Lumos$\mathcal{X}$ achieves SOTA identity similarity scores, demonstrating its strong ability to preserve identity consistency in video generation. The qualitative comparison shown in Fig. 5 (a) further highlights the advanced performance of our Lumos$\mathcal{X}$.

Table 1: Comparison of different methods for single-face identity-consistent video generation.

| Methods | Identity Consistency | | Prompt Following |
|---|---|---|---|
| | ArcSim ↑ | CurSim ↑ | ViCLIP-T ↑ |
| ConsisID (Yuan et al., 2024) | 0.458 | 0.474 | **0.263** |
| Concat-ID (Zhong et al., 2025) | 0.467 | 0.485 | 0.261 |
| **Lumos𝒳** | **0.542** | **0.575** | 0.262 |

Table 2: Comparison of different methods for identity-consistent video generation.

| Methods | Identity Consistency | | Prompt Following |
|---|---|---|---|
| | ArcSim ↑ | CurSim ↑ | ViCLIP-T ↑ |
| SkyReels-A2 (Fei et al., 2025) | 0.382 | 0.401 | 0.261 |
| Phantom (Liu et al., 2025) | 0.508 | 0.536 | **0.264** |
| **Lumos𝒳** | **0.510** | **0.540** | 0.262 |

Table 3: Comparison of different methods for subject-consistent video generation, including evaluation of the entire video and evaluation on the extracted subjects.

| Methods | Entire Video | | | Extracted Subjects | | | | |
|---|---|---|---|---|---|---|---|---|
| | Dynamic ↑ | ViCLIP-T ↑ | ViCLIP-V ↑ | CLIP-T ↑ | CLIP-I ↑ | DINO-I ↑ | ArcSim ↑ | CurSim ↑ |
| SkyReels-A2 (Fei et al., 2025) | 0.671 | 0.251 | 0.839 | 0.178 | 0.606 | 0.192 | 0.271 | 0.290 |
| Phantom (Liu et al., 2025) | 0.661 | 0.254 | 0.865 | 0.185 | 0.647 | 0.216 | 0.444 | 0.477 |
| **Lumos𝒳** | **0.723** | **0.260** | **0.932** | **0.201** | **0.692** | **0.261** | **0.454** | **0.483** |

**Subject-consistent video generation.** This experiment focuses on comprehensive multi-subject video customization, with reference inputs including faces, attributes, objects, and background images. For the evaluation of the entire video, we adopt ViCLIP-T and ViCLIP-V to assess the semantic similarity, while Dynamics detects potential copy-paste artifacts in the motion consistency. For evaluation on subjects, CLIP-T, CLIP-I, and DINO-I measure the semantic similarity, while ArcSim and CurSim evaluate facial similarity within the extracted subject regions, collectively capturing the accuracy of face-attribute associations in multi-subject video generation. As shown in Tab. 3, our method achieves SOTA performance in both the entire video generation quality and the accuracy of face-attribute associations within the subject regions, outperforming the advanced models SkyReels-A2 (Fei et al., 2025) and Phantom (Liu et al., 2025), thus demonstrating the robustness of our Lumos𝒳. For qualitative comparison, Fig. 5(b) shows that our method supports flexible multi-subject foreground and background customization while maintaining accurate face–attribute matching across multiple subjects. In contrast, competing methods exhibit incorrect face–attribute pairings at a relatively high frequency, such as the mismatches visible in the lower-left SkyReels-A2 and Phantom results and the lower-right SkyReels-A2 result in Fig. 5(b), which further underscores the robustness of our approach.

## 4.3 ABLATION STUDY

In this paper, we focus on enabling flexible foreground-background customization while addressing the dependency of face-attribute across multiple subjects. Therefore, we conduct the ablation study of our Lumos𝒳 on subject-consistent video generation. We conduct experiments under a relatively lightweight setting, where the training dataset consists of 300K video samples while maintaining the same subject distribution as the full dataset, and the video generation resolution is set to 240p.

Table 4: Ablation study for the subject-consistent video generation. The value in the MACM parentheses is $r$.

| Methods | CLIP-T ↑ | ArcSim ↑ |
|---|---|---|
| None | 0.184 | 0.316 |
| +R2PE | 0.178 | 0.363 |
| +R2PE + CSAM | 0.182 | 0.363 |
| +R2PE + CSAM + MCAM (0.1) | 0.182 | 0.364 |
| +R2PE + CSAM + MCAM (0.5) | 0.186 | **0.429** |
| +R2PE + CSAM + MCAM (1.0) | **0.187** | 0.384 |

We ablate each component in Lumos𝒳, with the results presented in Tab. 4. The results show that R2PE (Row 2) significantly improves ArcSim, thanks to the binding of faces and attributes during the positional encoding, which helps the model avoid face confusion in generation. But CLIP-T shows a slight decrease, which we believe may be due to the shared T-idx within the subject group, affecting the semantic representation of individual entities. Furthermore, our CSAM (Row 3) shows an improvement over CLIP-T, as it effectively blocks the interaction between different conditional signals and allows the denoising branch to independently aggregate these signals. For MCAM (Rows 4–6), we evaluated performance under different values of the hyperparameter $r$. The results indicate that MCAM yields substantial improvements in both CLIP-T and ArcSim, as it not only enhances the semantic representation of each visual condition but also effectively optimizes intra-group and inter-group correlations within subject groups. ArcSim achieves its best performance at $r = 0.5$ (Row 5), while CLIP-T peaks at $r = 1.0$ (Row 6). Since the improvement in ArcSim is more significant at $r = 0.5$, and considering that ArcSim better reflects the accuracy of face-attribute affiliation matching, we ultimately choose $r = 0.5$ in Lumos𝒳.

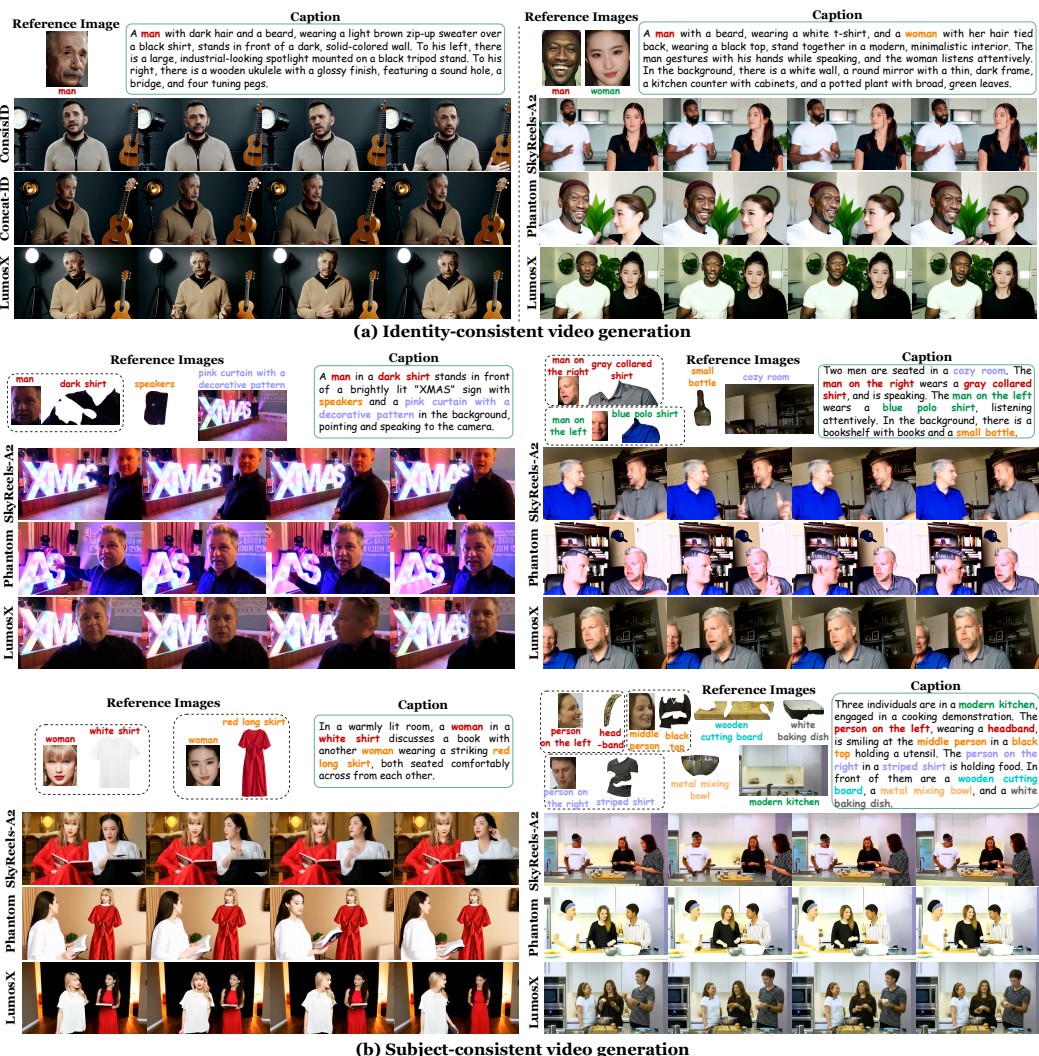

Figure 5: Qualitative comparison for identity-consistent and subject-consistent video generation.

## 5 CONCLUSION

This paper proposes Lumos$\mathcal{X}$, a novel framework designed for personalized multi-subject video generation that explicitly models face-attribute dependencies. To solve the lack of annotated data tailored for multi-subject generation, we develop a data collection pipeline that supports open-set entities with subject-specific dependencies. Built upon Wan2.1's T2V model, Lumos$\mathcal{X}$ introduces Relational Self-Attention and Relational Cross-Attention, which incorporate position embedding and attention mechanisms to explicitly bind face-attribute pairs into coherent subject groups and optimize both intra-group and inter-group correlations. Extensive experiments validate the effectiveness of Lumos$\mathcal{X}$ in generating fine-grained and personalized multi-subject videos, achieving state-of-the-art performance across diverse benchmarks.

## ACKNOWLEDGMENTS

This work was supported by the State Key Laboratory of Industrial Control Technology, China (Grant No. ICT2024A09).

ETHICS STATEMENT

Our study and dataset construction conform to established ethical standards, with no direct involvement of human subjects and no foreseeable risk of harm. Data usage complies with privacy and legal requirements, and we have aimed to mitigate potential biases in annotations and model evaluation. We disclose no conflicts of interest or sponsorship that could influence the results.

REPRODUCIBILITY STATEMENT

We have thoroughly described all proposed models and algorithms, experimental configurations, and the benchmarks used in our experiments in the main text and the appendix. Furthermore, we confirm that the code for this work will be released upon acceptance.

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

# Appendix of Lumos$\mathcal{X}$

In this Appendix, we provide additional content organized as follows:

- **Sec. A** details the **data collection pipeline and training dataset**, including:
  - Sec. A.1 Entity words retrieval and subject–attributes matching.
  - Sec. A.2 Obtaining condition images.
  - Sec. A.3 Data cleaning and filtering for training dataset.
  - Sec. A.4 Robustness analysis of external modules in the data collection pipeline.
- **Sec. B** provides additional details on **Lumos$\mathcal{X}$**, including:
  - Sec. B.1 Training objective.
  - Sec. B.2 Data augmentation details during training.
  - Sec. B.3 Generalization of Lumos$\mathcal{X}$ to diverse T2V architectures.
- **Sec. C** presents **extended ablation studies and evaluations**, including:
  - Sec. C.1 Fine-grained quantitative comparisons of identity-consistent video generation.
  - Sec. C.2 Fine-grained quantitative comparison of subject-consistent video generation.
  - Sec. C.3 Additional quantitative results from various online sources.
  - Sec. C.4 Discussion on Lumos$\mathcal{X}$'s capability for customized control for 4+ subjects.
  - Sec. C.5 Quantitative assessment of temporal coherence among different methods.
  - Sec. C.6 Evaluation on public personalization benchmark.
  - Sec. C.7 Visual effectiveness of individual components of Lumos$\mathcal{X}$.
  - Sec. C.8 Qualitative comparison under varying hyperparameters $r$ in MCAM.
  - Sec. C.9 Quantitative analysis of text-based attribute control.
  - Sec. C.10 Quantitative comparison with image-personalization–based initialization.
  - Sec. C.11 Discussion of the importance of the inpainting model in data collection pipeline.
  - Sec. C.12 Human study for multi-subject video customization.
  - Sec. C.13 Analysis of computational overhead and latency in Lumos$\mathcal{X}$.
- **Sec. D** provides **more visualization results**, including:
  - Sec. D.1 Additional results of identity-consistent video generation.
  - Sec. D.2 Additional results of subject-consistent video generation.
- **Sec. E** discusses the **limitations and future work**.
- **Sec. F** provides the **LLM usage statement**.
- **Sec. G** addresses the **broader impact** of our approach.

## A    DETAILS OF DATA COLLECTION PIPELINE AND TRAINING DATASET

### A.1    ENTITY WORDS RETRIEVAL AND SUBJECT-ATTRIBUTES MATCHING

In Sec. 3.2 of the manuscript, we propose utilizing Qwen2.5-VL-32B (Bai et al., 2025) to retrieve multiple entity words from the caption, while leveraging prior visual information from human detection results to achieve precise face-attribute matching. The process of processing a human-detecting frame with the corresponding caption can be divided into four steps, as shown in Fig. 6. The result of Data Organization (step 2) serves as the input to Qwen2.5-VL and includes: captions (text), bounding boxes (text), and frames with colored bounding boxes (visual). During the Data Processing (third step) phase, we design a tailored prompt for Qwen2.5-VL to better address the task requirements, as shown below. The results of Data Processing, as shown in the fourth column ("Data Annotation") of Fig. 6, involve extracting key entity words from the captions and categorizing them into *Subjects*, *Objects*, and *Background*. For Subjects, we obtain human-attribute dependency relations, which strictly follow the matching structure presented in the human-detecting frame.

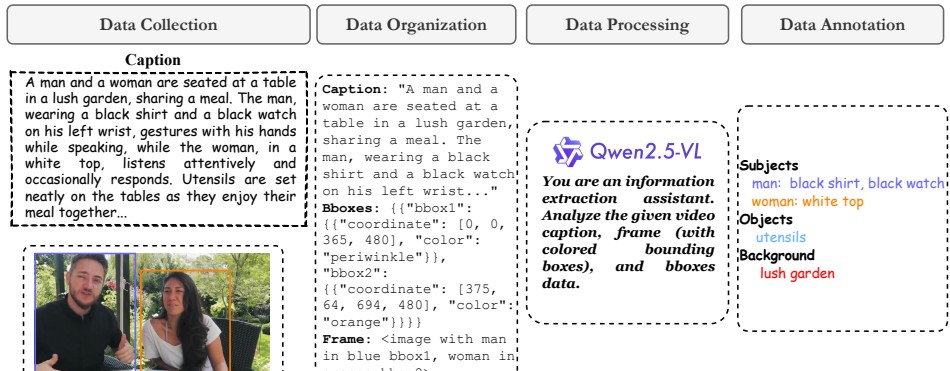

Figure 6: Process of entity words retrieval and subject-attributes matching with four steps: Data Collection, Data Organization, Data Processing, and Data Annotation.

Prompt design for entity words retrieval and subject-attribute matching in Qwen2.5-VL

```
PROMPT =
"""
You are an information extraction assistant. Analyze the given video caption, frame (
    with colored bounding boxes), and bounding box data.
Rules:
1. For PERSONS:
- Identify how many persons by the caption and bboxes data, list ALL human-related
    terms (human/person/man/woman/girl/boy etc.). Merge the same person to avoid
    redundancy.
- Extract ONLY concrete attributes for each person:
  [OK] Pick maximum THREE most frequent concrete attributes.
  [X] The attibutes can be clothing descriptions (e.g., "black suit", "red dress", "
      glasses", "hat") or physical features (e.g., "blonde hair", "beard")
  [X] REJECT abstract attributes (e.g., "skin tone", emotions/actions like "smiling"
      or "running")
  [X] REJECT additional descriptions of the attributes (e.g., red hoodie with a black
      design on the front -> red hoodie)
  [X] REJECT long phrases: keep under 5 words per attribute
- Crucially, use the Frame and BBoxes input: Visually match each person described in
    the Caption to the person inside the corresponding colored bbox (red, green, blue)
    depicted in the Frame. The BBoxes variable maps color names (e.g., "red") to bbox
    identifiers (e.g., "bbox1"). Assign the correct bbox identifier to the matched
    person to prevent mismatches.

2. For OTHER SUBJECTS:
[OK] Pick maximum THREE most frequent concrete objects
[OK] Use exact object names (e.g., "dog", "oak tree", "wooden chair")
[X] REJECT objects that cannot be directly described with specific nouns (e.g., small
    rectangular object)
[X] REJECT words that may refer to multiple objects (e.g., "leaves")
[X] REJECT long phrases: keep under 5 words per attribute

3. For BACKGROUND:
Nouns indicating the overall environment or backdrop of the video
[OK] Extract the exact environmental phrase (e.g., "park", "coffee shop")
[X] REJECT inferred locations

4. NOTES:
- The extracted phrases must strictly come from the caption, without adding, removing,
    or rewriting words and punctuation marks.
- Ensure bbox references correspond to the correct colored bbox from the input.
- Bbox coordinate format: [xmin, ymin, xmax, ymax].
"""
```

## A.2 OBTAINING CONDITION IMAGES

The detailed process for obtaining condition images, described in Sec. 3.2, is illustrated in Fig. 7. For subjects, we first apply face detection (Wang et al., 2024a) within the bounding boxes obtained from human detection to extract facial crops. Then, guided by the extracted word tags from Qwen2.5-VL (Bai et al., 2025), we use SAM (Kirillov et al., 2023) to segment the corresponding attribute

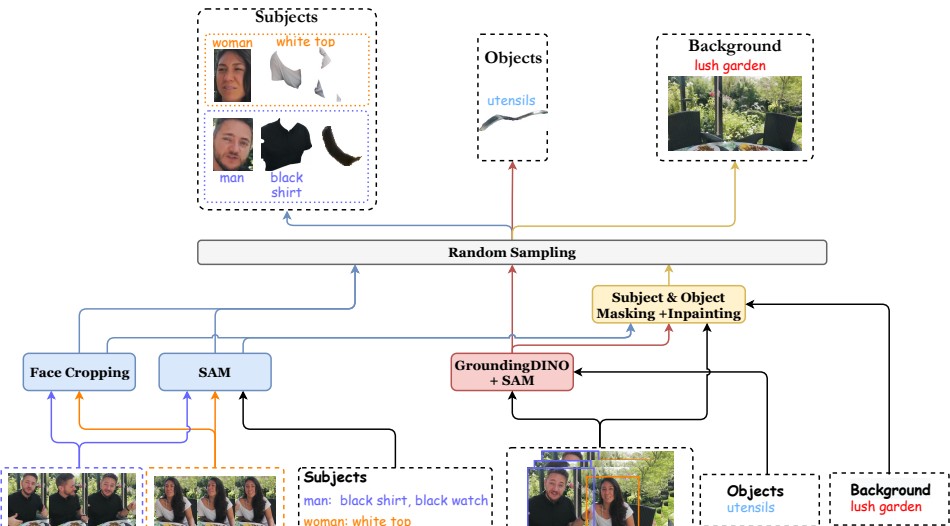

Figure 7: Process of acquiring condition images of subjects, objects, and the background.

regions within these human bounding boxes. For objects, we first employ GroundingDINO (Liu et al., 2024), guided by object tags, to detect bounding boxes from the global image. We then apply SAM within these bounding boxes to segment the mask regions corresponding to each object entity. As for the background, we first remove the subjects and objects based on the previously obtained crops and masks in face cropping and SAM. Since SAM occasionally produces imprecise boundaries, we dilate the foreground masks before inpainting. We then utilize the diffusion-based inpainting model FLUX (Labs, 2023), guided by the prompt: *"{background}, empty, nothing, there is nothing,"* where {*background*} corresponds to the extracted background tag. Finally, based on the valid outputs from the three key frames, we randomly select one result for each entity to serve as its corresponding condition image. This design aligns with the inference process, where each condition typically corresponds to a single reference image. Additionally, the random selection prevents all condition images from originating from the same frame, thereby enhancing data diversity.

## A.3 DATA CLEANING AND FILTERING FOR TRAINING DATASET

Our personalized multi-subject video generation training dataset is constructed based on Panda70M (Chen et al., 2024). However, not all data in the Panda70M dataset meets the quality standards required for video generation training. To ensure the suitability of the dataset, we perform a filtering and cleaning process based on the following criteria and operations:

- Removing subtitles using a text detection pipeline based on PaddleOCR (Baidu, 2025).
- Cropping black-and-white borders using a simple Hough transform-based approach.
- Excluding grayscale or black-and-white videos to ensure sufficient visual quality and aesthetic appeal.
- Retaining only samples with QAlign quality $> 3.5$ (Wu et al., 2023) and aesthetic score $> 2.0$ (Wu et al., 2023), enforcing constraints on semantic alignment and visual aesthetics.
- Retaining only samples with flow strength within the range of 0.05 to 2.0, constraining motion intensity to an appropriate level for video generation.
- Retaining only videos in which 1 to 3 people are detected by Yolov9 (Wang et al., 2024a), as videos with more than 3 individuals typically exhibit lower visual and semantic quality.
- Removing duplicate videos by clustering based on VideoCLIP (Wang et al., 2024b) embeddings.

After data cleaning and filtering, we collect a total of 1.57M video samples, comprising 1.31M single-subject, 0.23M two-subject, and 0.03M three-subject videos.

### A.4 ROBUSTNESS ANALYSIS OF EXTERNAL MODULES IN THE DATA COLLECTION PIPELINE

To ensure the quality and robustness of the data collection pipeline, we adopt deliberate component choices and strict filtering strategies to reduce upstream prediction errors:

- *Careful Selection of High-Reliability Components*: 1) For face–attribute tag extraction, we used Qwen-VL-32B, which incorporates visual priors, instead of the pure language model Qwen-2.5-32B, improving accuracy (see Sec. A); 2) For background inpainting, we selected FLUX over Stable Diffusion 2.0 due to its superior realism.

- *Strict Data Filtering Policies*: 1) Qwen-VL predictions were accepted only if the predicted entities appeared in the captions; 2) Conservative thresholds were used in detection modules (`box_threshold=0.5`, `text_threshold=0.45` in Grounding DINO and SAM) to eliminate unreliable region-text matches; 3) For background inpainting, we retained only samples where the foreground occupied less than 50% of the image to maintain consistency and plausibility. Additional filters are detailed in Sec. A.3.

While isolating the robustness of each external module is computationally expensive, we conduct a simple test of Qwen's robustness in word tag extraction. A case was marked as incorrect if either the face or attribute word was missing from the caption, or if the pairing was semantically invalid. With visual priors, Qwen-VL-32B achieved 95.2% accuracy, compared to 78.4% for Qwen-2.5-32B without visual input. Overall, our stringent quality-driven filtering is evident in the data selection ratio: from 70 million Panda70M samples, only 1.57 million videos were retained (2.2%). This highlights our strong emphasis on quality control and minimizing error propagation.

## B DETAILS AND DISCUSSIONS OF LUMOS$\mathcal{X}$

### B.1 TRAINING OBJECTIVE

Due to our Lumos$\mathcal{X}$ built upon Wan2.1's (Wang et al., 2025) T2V model, we adopt Flow Matching Lipman et al. (2022) to formulate the training objective. Flow Matching is a generative paradigm that learns continuous-time dynamics using ordinary differential equations (ODEs). It directly estimates optimal transport paths between distributions, enabling stable training without iterative denoising steps. Given a video latent representation $\mathbf{z}$, a random noise $\mathbf{z}_0$, and a timestep $t \in [0, 1]$ drawn from a logit-normal distribution, an intermediate latent $\mathbf{z}_t$ is obtained as the training input. Following Rectified Flows (RF) (Esser et al., 2024), the intermediate state $\mathbf{z}_t$ is defined via linear interpolation between $\mathbf{z}$ and $\mathbf{z}_0$, formulated as:

$$\mathbf{z}_t = (1 - t) \cdot \mathbf{z}_0 + t \cdot \mathbf{z}, \tag{6}$$

The corresponding ground-truth velocity is given by the time derivative of $x_t$:

$$\mathbf{v}_t = \frac{\mathrm{d}\mathbf{z}_t}{\mathrm{d}t} = \mathbf{z} - \mathbf{z}_0. \tag{7}$$

The model is trained to estimate this velocity field. The training objective is defined as the Mean Squared Error (MSE) between the predicted velocity and the ground-truth velocity:

$$\mathcal{L} = \mathbb{E}_{\mathbf{z}_0, \mathbf{z}, \mathbf{c}_{text}, \mathbf{z}_c, t} \left\| u(\mathbf{z}_t, \mathbf{c}_{text}, \mathbf{z}_c, t; \theta) - \mathbf{v}_t \right\|^2, \tag{8}$$

where $\mathbf{c}_{text}$ and $\mathbf{z}_c$ denote the umT5 text embedding and reference visual embedding, respectively. $\theta$ is the model weights, and $u(\mathbf{z}_t, \mathbf{c}_{text}, \mathbf{z}_c, t; \theta)$ denotes the output velocity predicted by the generation model.

### B.2 DATA AUGMENTATION DETAILS DURING TRAINING

During training, all conditional reference images are resized to match the video aspect ratio. Subject and object reference images are augmented with both numerical transformations (*e.g.*, brightness, blur) and geometric transformations (*e.g.*, rotation, horizontal flip), while background images are augmented only with numerical augmentations.

## B.3 Generalization of Lumos𝒳 to Diverse T2V Architectures

While our current implementation is based on Wan2.1-T2V with a single-tower DiT architecture, the proposed modules in Lumos𝒳 are architecture-agnostic and compatible with other DiT-style backbones, such as the dual-tower MM-DiT in HunyuanVideo (Kong et al., 2024) and the Parallel Attention in MAGI-1 (Sand-AI, 2025). Specifically, R2PE simply reorders relative position indices and can be seamlessly integrated into any DiT-based model, as it is independent of tower design. CSAM and MCAM function as attention masks for Self-Attention (among visual tokens) and Cross-Attention (between visual and textual tokens), respectively. In MM-DiT, these operations occur after dual-stream fusion and can adopt the same masking logic. Similarly, Parallel Attention includes equivalent attention interfaces that support our mask-based modules directly. In essence, regardless of whether the base model is Wan2.1, HunyuanVideo, or MAGI-1, all DiT-based T2V architectures share two key components: intra-modal interactions among visual tokens and cross-modal attention between visual and textual tokens. Our modules are explicitly designed to operate at these points, enabling straightforward extension to diverse architectures.

## C More Ablation Discussion

### C.1 Fine-Grained Quantitative Comparisons of Identity-Consistent Video Generation

We have added comparisons with Phantom and SkyReels-A2 in the single-face and multi-faces identity-consistent video generation setting, as shown in Tab 5. In the single-face setting (rows 1-3), our method underperforms Phantom in terms of identity preservation metrics (ArcSim and Cur-Sim). However, our Lumos𝒳 demonstrates a clear performance advantage in multi-face scenarios for identity preservation (rows 4–6).

### C.2 Fine-Grained Quantitative Comparison of Subject-Consistent Video Generation

Table 5: Fine-grained quantitative comparison of identity-consistent video generation across different numbers of faces.

| Methods | Numbers | Identity Consistency | | Prompt Following |
|---|---|---|---|---|
| | | ArcSim ↑ | CurSim ↑ | ViCLIP-T ↑ |
| SkyReels-A2 (Fei et al., 2025) | 1 face | 0.509 | 0.540 | 0.258 |
| Phantom (Liu et al., 2025) | 1 face | **0.602** | **0.637** | 0.261 |
| **Lumos𝒳** | 1 face | 0.542 | 0.575 | **0.262** |
| SkyReels-A2 (Fei et al., 2025) | ≥ 2 faces | 0.282 | 0.292 | 0.263 |
| Phantom (Liu et al., 2025) | ≥ 2 faces | 0.434 | 0.457 | **0.266** |
| **Lumos𝒳** | ≥ 2 faces | **0.485** | **0.513** | 0.263 |

To further demonstrate the superiority of our model in handling multi-subject settings, we conduct a more fine-grained quantitative evaluation of subject-consistent video generation across varying numbers of subjects (1-3 subjects), as shown in Tab. 6. It can be observed that as the number of subjects increases, Lumos𝒳 demonstrates increasingly superior performance across the majority of metrics, with the performance gap over other methods becoming more pronounced. These experimental results further validate the effectiveness of the module we designed in Lumos𝒳 for modeling multi-subject face-attribute dependency relationships.

### C.3 Additional Quantitative Results from Various Online Sources

We also provide additional qualitative results using condition images collected from various online sources. The test includes 50 sample cases (20 single-subject, 20 two-subject, and 10 three-subject cases). Since ground truth videos are not available for these samples, ViCLIP-V cannot be measured. As shown in Tab. 7, our method continues to outperform other approaches on most metrics, particularly those related to subject-specific customization (CLIP-T, CLIP-I, DINO-I, ArcSim, Cur-Sim).

Table 6: Fine-grained quantitative comparison of subject-consistent video generation across different numbers of subjects. (·) denotes that the respective method yields a lower value than our Lumos$\mathcal{X}$ on this metric, whereas (·) indicates better performance. Values in parentheses are recalibrated differences computed as (**Method** − **Lumos$\mathcal{X}$**) for the same setting and metric.

| Methods | Numbers | Entire Video | | | Extracted Subjects | | | | |
|---|---|---|---|---|---|---|---|---|---|
| | | Dynamic ↑ | ViCLIP-T ↑ | ViCLIP-V↑ | CLIP-T ↑ | CLIP-I ↑ | DINO ↑ | ArcSim ↑ | CurSim ↑ |
| SkyReels-A2 | 1 subject | 0.772 (0.080) | 0.252 (0.007) | 0.870 (0.064) | 0.177 (0.022) | 0.604 (0.082) | 0.199 (0.069) | 0.356 (0.133) | 0.380 (0.136) |
| Phantom | 1 subject | 0.848 (0.004) | 0.254 (0.005) | 0.880 (0.054) | 0.186 (0.013) | 0.646 (0.040) | 0.215 (0.053) | **0.539** (0.050) | **0.581** (0.065) |
| **Lumos$\mathcal{X}$** | 1 subject | **0.852** | **0.259** | **0.934** | **0.199** | **0.686** | **0.268** | 0.489 | 0.516 |
| SkyReels-A2 | 2 subjects | 0.655 (0.009) | 0.251 (0.011) | 0.820 (0.110) | 0.182 (0.021) | 0.620 (0.079) | 0.185 (0.064) | 0.216 (0.227) | 0.233 (0.241) |
| Phantom | 2 subjects | 0.526 (0.138) | 0.255 (0.007) | 0.856 (0.074) | 0.188 (0.015) | 0.658 (0.041) | 0.215 (0.034) | 0.396 (0.047) | 0.424 (0.050) |
| **Lumos$\mathcal{X}$** | 2 subjects | **0.664** | **0.262** | **0.930** | **0.203** | **0.699** | **0.249** | **0.443** | **0.474** |
| SkyReels-A2 | 3 subjects | 0.268 (0.156) | 0.243 (0.017) | 0.795 (0.141) | 0.161 (0.039) | 0.546 (0.144) | 0.189 (0.097) | 0.150 (0.202) | 0.161 (0.219) |
| Phantom | 3 subjects | 0.411 (0.013) | 0.250 (0.010) | 0.840 (0.096) | 0.169 (0.031) | 0.601 (0.089) | 0.220 (0.066) | 0.247 (0.105) | 0.269 (0.111) |
| **Lumos$\mathcal{X}$** | 3 subjects | **0.424** | **0.260** | **0.936** | **0.200** | **0.690** | **0.286** | **0.352** | **0.380** |
| SkyReels-A2 | 4 subjects | 0.257 (0.047) | 0.241 (0.022) | 0.761 (0.165) | 0.111 (0.095) | 0.375 (0.329) | 0.116 (0.149) | 0.122 (0.167) | 0.129 (0.190) |
| Phantom | 4 subjects | 0.286 (0.018) | 0.240 (0.023) | 0.790 (0.136) | 0.165 (0.041) | 0.615 (0.089) | 0.200 (0.065) | 0.191 (0.098) | 0.200 (0.119) |
| **Lumos$\mathcal{X}$** | 4 subjects | **0.304** | **0.263** | **0.926** | **0.206** | **0.704** | **0.265** | **0.289** | **0.319** |

Table 7: Comparison of different methods for subject-consistent video generation, including evaluation of the entire video and evaluation on subjects.

| Methods | Entire Video | | Extracted Subjects | | | | |
|---|---|---|---|---|---|---|---|
| | Dynamic ↑ | ViCLIP-T ↑ | CLIP-T ↑ | CLIP-I ↑ | DINO-I ↑ | ArcSim ↑ | CurSim ↑ |
| SkyReels-A2 (Fei et al., 2025) | 0.839 | 0.235 | 0.207 | 0.614 | 0.173 | 0.160 | 0.166 |
| Phantom (Liu et al., 2025) | **0.929** | **0.255** | 0.226 | 0.623 | 0.195 | 0.271 | 0.305 |
| **Lumos$\mathcal{X}$** | 0.804 | 0.247 | **0.232** | **0.668** | **0.336** | **0.317** | **0.346** |

**Caption**: "**10 people** looking directly at the camera with fully visible frontal faces. They toss and catch glowing objects between each other in a high-tech arena. Fast coordinated motion, but faces always unobstructed and well-lit."

**Wan2.1-T2v-1.3B**

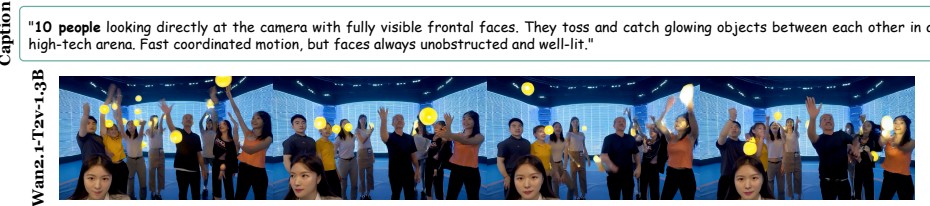

Figure 8: Visualization of 10-person T2V results of our baseline model Wan2.1-T2V-1.3B.

## C.4 DISCUSSION ON LUMOS$\mathcal{X}$'S CAPABILITY FOR CUSTOMIZED CONTROL FOR 4+ SUBJECTS

Lumos$\mathcal{X}$ is designed with scalability in mind. Although the training dataset contains videos with up to three subjects, the model architecture, including R2PE, CSAM, and MCAM, is inherently scalable and capable of handling more subjects during inference without retraining. To validate this capability, we conduct quantitative comparisons under a four-subject setting using 50 videos, without any retraining. The results in Tab. 6 show that even in the four-subject setting, our method consistently outperforms the baselines. Moreover, performance remains stable compared to the three-subject setting, showing no significant drop. It is worth noting that although R2PE in Lumos$\mathcal{X}$ inherits the extrapolation capability of RoPE, increasing the number of subjects slightly beyond what was seen during training (*e.g.*, from 3 to 4) does not lead to a noticeable drop in performance during inference without re-training. However, when the subject count increases substantially (*e.g.* from 3 to 10+), the risk of extrapolation instability rises significantly: higher RoPE dimensions may not have seen a full rotation period during training (Liu et al., 2023), causing positional encodings to become Out-of-Distribution (O.O.D), which may degrade attention alignment. To support long-range inference without retraining, we plan to try to integrate NTK-Aware Scaled RoPE (NTK-RoPE) (Liu et al., 2023), a training-free length extrapolation method that extends RoPE's context range by adjusting the base of its sinusoidal encoding rather than learning new positional parameters. This change enables the model to handle positions beyond the training window with minimal perplexity degradation and no fine-tuning required. In addition to RoPE extrapolation, we also find that the generation capability of the underlying base model itself plays a crucial role in determining the upper limit of how many subjects the customization model can realistically handle. For example, Wan2.1-1.3B-T2V, which we rely on, also struggles significantly when generating videos containing more than ten distinct human subjects. Fig. 8 presents visualizations of Wan2.1-1.3B-T2V under a

Table 8: Quantitative comparison of temporal coherence metrics on subject-consistent video generation.

| Methods | Subject Consistency ↑ | Background Consistency ↑ | Motion Smoothness ↑ | Face Consistency ↑ |
|---|---|---|---|---|
| SkyReels-A2 (Fei et al., 2025) | 0.654 | 0.798 | 0.979 | 0.739 |
| Phantom (Liu et al., 2025) | 0.768 | 0.853 | 0.986 | 0.883 |
| **Lumos$\mathcal{X}$** | **0.962** | **0.946** | **0.988** | **0.895** |

10-subject setting, where the facial quality is severely degraded and the model fails to strictly follow the prompt, generating only nine people instead of ten. This reinforces our point that the base model's capability is the primary bottleneck when extending Lumos$\mathcal{X}$ to scenarios involving substantially more subjects. These results indicate that current video-generation models like Wan-2.1 lack the capacity to learn stable representations for such large numbers of subjects, largely because high-quality 10+ subject videos are extremely rare.

## C.5 QUANTITATIVE ASSESSMENT OF TEMPORAL COHERENCE AMONG DIFFERENT METHODS

To evaluate temporal coherence, we additionally report four metrics: Subject Consistency, Background Consistency, Motion Smoothness, and Face Consistency. The first three are adopted from VBench (Huang et al., 2024). Subject Consistency measures how consistently the subject (*e.g.*, person or object) appears across consecutive frames. Background Consistency evaluates the temporal stability of the background, while Motion Smoothness quantifies the fluidity and natural continuity of motion. In contrast, Face Consistency is a metric we define to capture frame-to-frame facial similarity. Specifically, we use ArcFace embeddings to compute the similarity of the same face across consecutive frames. The results are shown in Tab. 8, evaluated on the test set for subject-consistent video generation, which contains 500 videos. Our Lumos$\mathcal{X}$ outperforms other methods across all metrics.

## C.6 EVALUATION ON PUBLIC PERSONALIZATION BENCHMARK

To further evaluate our method, we conducted a quantitative comparison between our Lumos$\mathcal{X}$ and other approaches, including SkyReels-A2 and Phantom, on the MSRVTT-personalization (Chen et al., 2025) benchmark.

Table 9: Quantitative comparison on MSRVTT-personalization (Chen et al., 2025) benchmark. **boldfacen** and underline font indicate the highest and the second highest results.

| Method | #Base Model | Text-S ↑ | Vid-S↑ | Subj-S↑ | Dync-D↑ |
|---|---|---|---|---|---|
| SkyReels-A2 (Fei et al., 2025) | Wan2.1-14B-T2V | 0.253 | **0.781** | **0.554** | 0.783 |
| Phantom (Liu et al., 2025) | Wan2.1-1.3B-T2V | **0.270** | 0.696 | 0.534 | 0.461 |
| **Lumos$\mathcal{X}$** | Wan2.1-1.3B-T2V | 0.258 | 0.707 | 0.549 | **0.786** |

The MSRVTT-personalization benchmark provides two evaluation modes: subject-mode and face-mode. Face-mode focuses solely on face-similarity metrics, whereas we are interested in assessing the overall subject performance (face + attributes). Therefore, we conduct our evaluation under the subject-mode setting. Our evaluation is conducted using one reference image for the subject and one for the background, and, the results are reported in Tab. 9. It is worth noting that MSRVTT-personalization is a single-subject benchmark, and its definition of 'subject' differs from ours: a subject may refer to either a human category (*e.g.*, man, woman) or an object category (*e.g.*, car, horse, clothes, hat). Moreover, for human subjects under the benchmark's subject-mode setting, the subject is provided as a holistic entity without decoupled face and attribute references, and therefore does not involve the face–attribute matching problem that our method is designed to address. As a result, our method does not benefit from its explicit face–attribute relational modeling in this setting. In addition, both Lumos$\mathcal{X}$ and Phantom are built on the Wan2.1-1.3B-T2V base model, whereas SkyReels-A2 adopts the much stronger Wan2.1-14B-T2V base model and therefore further benefits from a significantly more capable backbone. Despite these disadvantages, our Lumos$\mathcal{X}$ still achieves the second-best overall performance, demonstrating the robustness and generalization ability of our approach even outside its primary multi-subject setting.

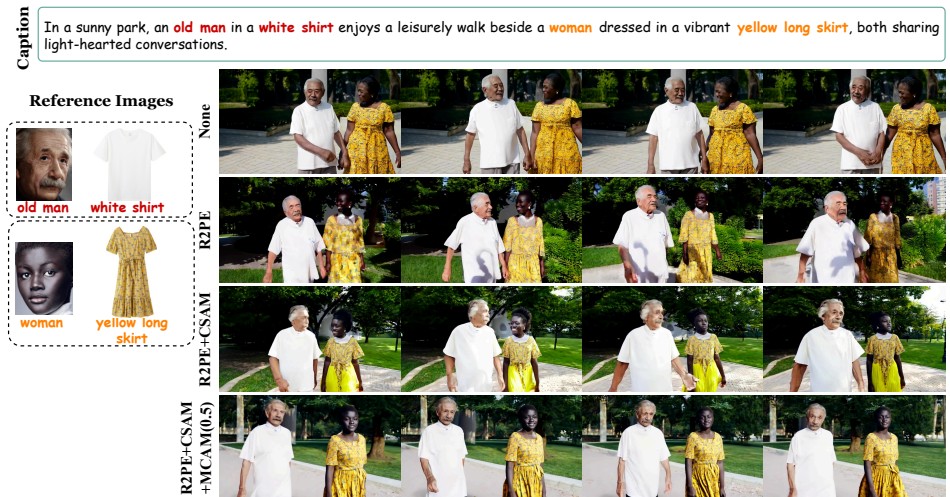

Figure 9: Visual effectiveness of individual components proposed in our Lumos$\mathcal{X}$.

## C.7 VISUAL EFFECTIVENESS OF INDIVIDUAL COMPONENTS OF LUMOS$\mathcal{X}$

To assess the impact of each proposed component in Lumos$\mathcal{X}$, we conduct a qualitative evaluation, as illustrated in Fig. 9. Without incorporating our proposed modules (first row), the generated video exhibits noticeable identity confusion. Specifically, an old white man is misrepresented as black, and a young black girl is erroneously depicted as old. With the introduction of our designed R2PE (second row), coarse-grained identity attributes, including age and skin tone, are effectively rectified. While the introduction of the CSAM (third row) further improves fine-grained facial identity preservation, certain artifacts still persist in the generated videos. Ultimately, the incorporation of the MCAM (fourth row) leads to improved identity consistency in the generated videos, while also enhancing video quality to a certain degree. To further validate the effectiveness of our CSAM and MCAM, we also visualize the Attention Similarity Scores averaged over all 12 heads in the last DiT layer for the case shown in Fig. 9. The results are presented in Fig. 10. It is worth noting that our generated videos contain 81 frames, and after VAE encoding, the temporal dimension is downsampled by a factor of four, resulting in a video noise token length of 21. The results in this figure can be compared with those in Fig. 4 in the main paper. From this comparison, we observe that CSAM (Fig. 10(a) *vs.* Fig. 10(b)) enables the denoising branch to independently aggregate conditional signals and to effectively bind face–attribute dependencies within the conditional branch. Moreover, MCAM (Fig. 10(c) *vs.* Fig. 10(d)) explicitly enhances intra-group and inter-group correlations among subject groups and strengthens the semantic-level representation of the visual condition tokens.

## C.8 QUALITATIVE COMPARISON UNDER VARYING HYPERPARAMETERS $r$ IN MCAM

When MCAM with $r = 0.1$ is introduced (third row), the artifacts in the generated videos are significantly alleviated. Nevertheless, identity consistency is still unsatisfactory; for instance, a short-sleeved T-shirt may appear as long-sleeved in the video, and facial similarity is still insufficient. At $r = 0.5$ (fourth row), we observe a notable improvement in identity consistency, especially in the resemblance between facial appearances in the generated videos and the reference image, along with a further enhancement in video quality. Increasing $r$ to 1.0 (fifth row) results in continued improvements in video quality, though it introduces a slight decline in identity similarity. The above findings align with the quantitative results reported in Sec. 4.3 of the manuscript. Accordingly, we choose $r = 0.5$ as the final setting based on a balanced trade-off between identity consistency and video quality.

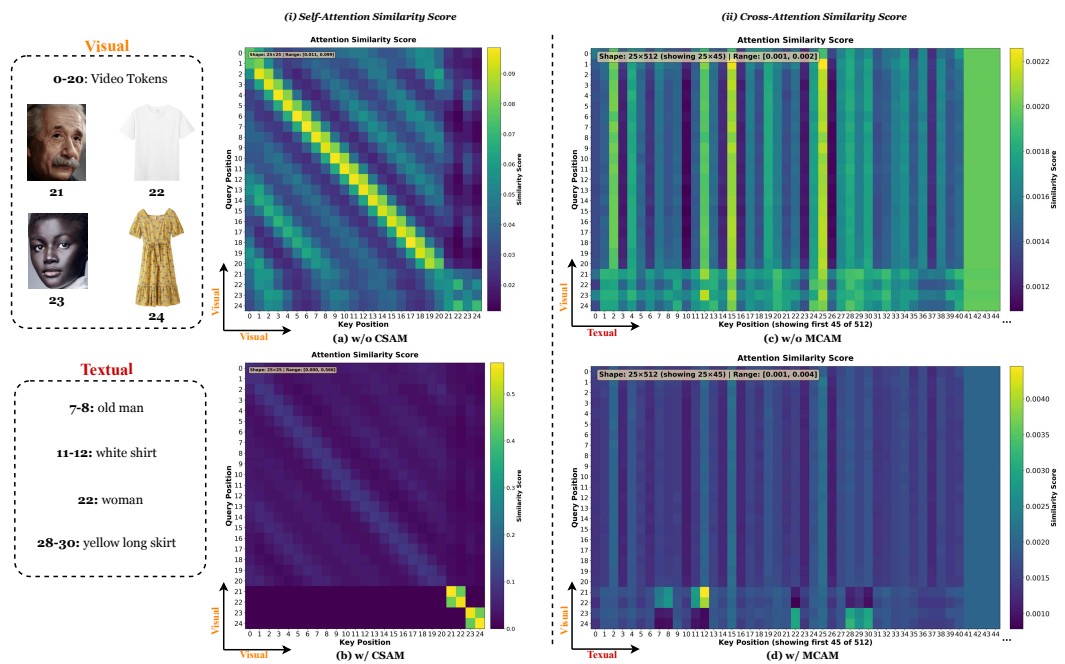

Figure 10: Visualization of attention maps. (i): Self-Attention similarity score. (ii): Cross-Attention similarity score.

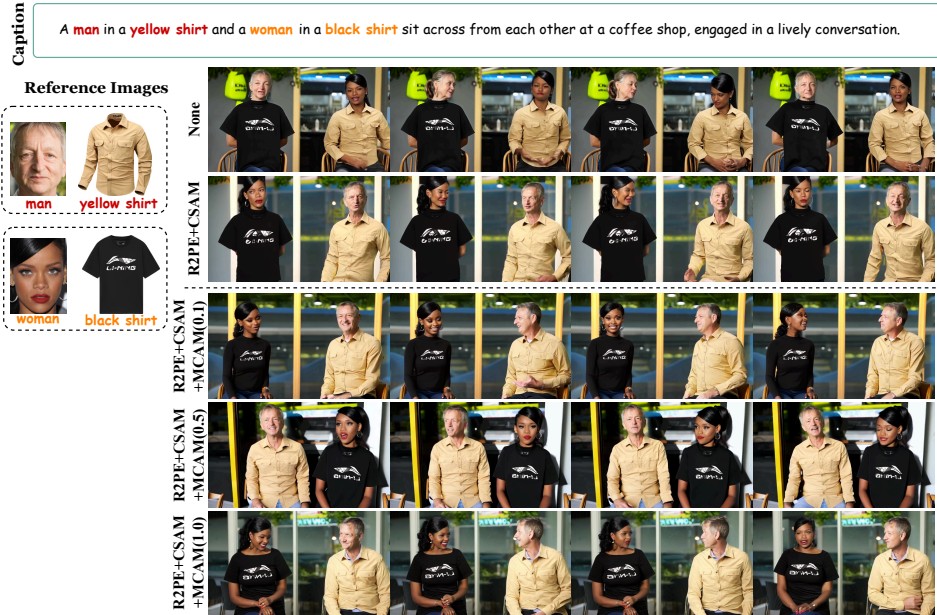

Figure 11: Qualitative comparison under varying control hyperparameters $r$ in MCAM.

## C.9 QUANTITATIVE ANALYSIS OF TEXT-BASED ATTRIBUTE CONTROL WITHOUT IMAGES IN MULTI-SUBJECT VIDEO CUSTOMIZATION

Our method supports attribute control using text alone, and this setting is already included in both the main paper and the appendix under Identity-Consistent Video Generation (Sec. 4.1, main paper). To more comprehensively evaluate this setting in multi-subject video customization scenarios, we perform subject-region quantitative evaluations under this setting, using the same metrics as in Subject-Consistent Video Generation, to compare text-based attribute control across different methods. The results are presented in the Tab. 11. Overall, the CLIP-T scores are higher under

text-only control, which is expected since the attributes are directly specified via text prompts, naturally yielding higher text–video similarity. CLIP-I trends largely mirror those of CLIP-T because both measure semantic similarity. Notably, for Lumos$\mathcal{X}$ (rows 6-7, col. 4), the performance gap between text-only control and text&visual control is very small, whereas for the other methods (rows 2-5, col. 4), performance under text&visual control drops noticeably compared to text-only control. This suggests that those methods become more prone to attribute confusion once visual conditions are introduced, likely due to the absence of explicit face–attribute relational modeling, whereas Lumos$\mathcal{X}$ avoids this issue. For DINO-I, text-only control is clearly weaker than text &visual control in Lumos$\mathcal{X}$. At the same time, the substantial improvement under text&visual control indicates that our face–attribute relational constraints are accurately matched and effectively utilized. In comparison, Phantom and SkyReels-A2 do not show notable DINO-I improvements when visual attribute conditions are added, suggesting that their face–attribute alignment is less reliable. Meanwhile, under the text-based attribute control setting, Lumos$\mathcal{X}$ performs on par with Phantom and clearly outperforms SkyReels-A2, demonstrating its strong generalization ability even without attribute images.

## C.10 Quantitative Comparison with Image-Personalization–Based Initialization

To evaluate whether multi-subject image personalization models can serve as an initial image generator prior to video synthesis, we conducted an additional experiment using UNO (Wu et al., 2025) to produce a multi-subject customized image, which was then fed into Wan2.1-14B-I2V (Wang et al., 2025) for video generation. We quantitatively compared this pipeline (UNO + Wan-I2V) against our method on our benchmark. The results are shown in Tab. 12. Overall, our method outperforms UNO + Wan2.1-I2V-14B, particularly on ViCLIP-V, DINO-I, FaceSim and CurSim, which reflects facial identity consistency. This improvement is expected, as UNO is not specifically designed for face-aware attribute binding, whereas our approach explicitly models face–attribute relational constraints, leading to significantly better identity preservation in video personalization. Our ViCLIP-T and CLIP-T scores are slightly weaker than those of UNO + Wan2.1-I2V-14B, which may be attributed to the latter's use of a substantially larger base model (Wan-14B).

## C.11 Discussion of the Importance of the Inpainting Model in Data Collection Pipeline

To quantitatively assess the realism of FLUX compared with other inpainting models, we conducted a large-scale evaluation on 2,130 test cases, applying both FLUX Labs (2023) and Stable-Diffusion-2 Rombach et al. (2022) for inpainting. We then computed FID scores against the COCO 2017 Val set to measure distributional realism. In addition, we randomly sampled 100 cases and asked GPT-4o to independently judge which inpainting result appeared more realistic. The instruction provided to GPT-4o is as follows:

---

**Instruction design for inpainting model assessment in GPT-4o**

```
PROMPT =
"""
- I will give you three images. The first two images are background inpainting results
    obtained by masking out the foreground in the original image and then applying two
    different inpainting models. The third image is the corresponding foreground mask.
    Please evaluate which of the first two images provides a better inpainting result
    based on pixel-level consistency, scene continuity, and the overall plausibility of
     the inpainted background. You should output only "1" or "2", indicating whether
    the first or the second image is better. Do not output anything else.

- Note that this is background completion, so the masked region should be inpainted
    with appropriate background content. Pay particular attention to the continuity and
     coherence of the background in the inpainted region.
"""
```

---

The results of both evaluations are summarized in the Tab. 10. The results show that FLUX achieves better performance under both evaluation metrics. To further understand how background quality affects downstream video generation, we performed qualitative comparison using the two inpainting

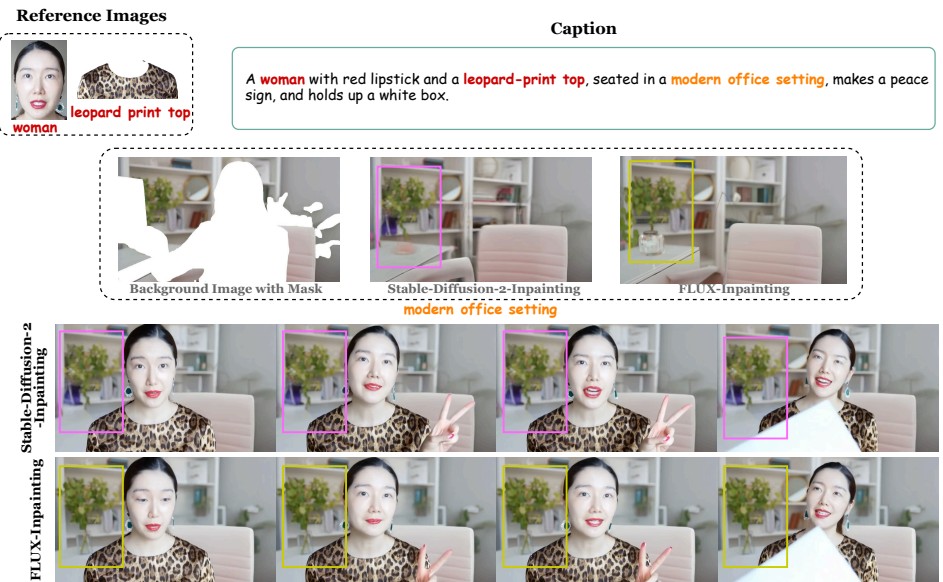

Figure 12: Qualitative comparison between Stable-Diffusion-2 and FLUX for background inpainting.

outputs as inputs to video generation. As shown in Fig. 12, artifacts introduced during background inpainting are clearly propagated into the generated video, degrading overall video quality. This confirms that inpainting realism plays a crucial role in high-quality video generation and motivates our choice of FLUX as the inpainting module.

## C.12 HUMAN STUDY FOR MULTI-SUBJECT VIDEO CUSTOMIZATION

To complement automated metrics, we conducted a user study on a randomly sampled set of 24 video cases, including 6 single-subject, 12 two-subject, and 6 three-subject customization scenarios. The comparison includes Lumos$\mathcal{X}$, SkyReels-A2, and Phantom, with a total of 30 participants. Each participant evaluated videos along four dimensions:

| Method | FID ↓ | AI Judgement ↑ |
|---|---|---|
| Stable-Diffusion-2-Inpainting | 96.32 | 36% |
| **FLUX-Inpainting** | **92.83** | **64%** |

Table 10: Comparison between Stable-Diffusion-2 Rombach et al. (2022) and FLUX Labs (2023) for background inpainting.

- **Face–Attribute Alignment**: whether each face is correctly matched with its corresponding attributes (*e.g.*, clothing, accessories, or hairstyle).

- **Face Similarity**: how closely each generated face resembles the provided visual reference.

- **Video Naturalness**: the overall visual quality and coherence of the generated video.

- **Prompt Adherence**: whether the generated video follows the instructions specified in the text prompt.

For each case, participants ranked the three methods, and scores were assigned as follows: 1 point for first place, 0.5 points for second place, and 0 points for third place. Final scores for each dimension were computed as the weighted average across all participants. The results are shown below Fig 13. Overall, our method achieves superior performance across all four evaluation dimensions.

## C.13 ANALYSIS OF COMPUTATIONAL OVERHEAD AND LATENCY IN LUMOS$\mathcal{X}$

To evaluate the computational impact of our relational attention mechanisms, we report detailed inference-stage statistics on computational overhead and latency. Specifically, we present the average latency for each Self-Attention and Cross-Attention operation, the average per-step latency, per-step FLOPs, and GPU memory consumption. All measurements are obtained on an H20 GPU

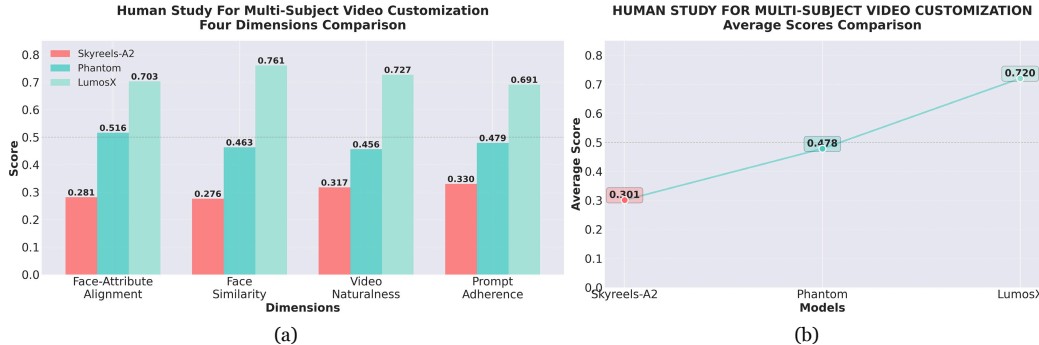

Figure 13: Human study results for multi-subject video customization. (a) Results across the four evaluation dimensions. (b) Average scores computed over all four dimensions.

Table 11: Quantitative comparison of text-only *vs.* text&visual attribute control across methods.

| Methods | Attribute Control | CLIP-T | CLIP-I | DINO-I |
|---|---|---|---|---|
| SkyReels-A2 (Fei et al., 2025) | text-only | 0.192 | 0.643 | 0.192 |
| | text&visual | 0.178 | 0.606 | 0.192 |
| Phantom (Liu et al., 2025) | text-only | 0.207 | 0.687 | 0.209 |
| | text&visual | 0.185 | 0.647 | 0.216 |
| **Lumos$\mathcal{X}$** | text-only | 0.205 | 0.684 | 0.210 |
| | text&visual | 0.193 | 0.681 | 0.265 |

Table 12: Comparison of image-personalization–based method and Lumos$\mathcal{X}$ for subject-consistent video generation, including evaluation of the entire video and evaluation on subjects.

| Methods | Entire Video | | | Extracted Subjects | | | | |
|---|---|---|---|---|---|---|---|---|
| | Dynamic ↑ | ViCLIP-T ↑ | ViCLIP-V↑ | CLIP-T ↑ | CLIP-I ↑ | DINO-I ↑ | ArcSim ↑ | CurSim ↑ |
| UNO + Wan2.1-I2V-14B | 0.547 | **0.261** | 0.880 | **0.201** | 0.650 | 0.197 | 0.237 | 0.244 |
| **Lumos$\mathcal{X}$** | **0.723** | 0.260 | **0.932** | **0.201** | **0.692** | **0.261** | **0.454** | **0.483** |

under the same video customization scenario. The results are shown in Tab. 13 and the key observations are summarized below:

- R2PE incurs no extra compute or memory overhead (row 1 *vs.* row 2), since it reorders relative position indices rather than introducing new parameters or operations.

- The original implementation of the Wan2.1-T2V model utilizes FlashAttention 2.0 for acceleration, which does not support custom masks. In our CSAM module, we replace it with MagiAttention (Sand-AI, 2025) (see Sec. 3.3.1 in the main paper), which supports custom masking and achieves faster inference (row 1 vs. row 3). The integration of CSAM results in only a modest overhead (row 4 *vs.* row 5).

- For MCAM (Cross-Attention), MagiAttention cannot handle numeric masks, so we use PyTorch's native implementation. Since the key comes from relatively short T5 token sequences (512 tokens), the computation remains lightweight. As a result, MCAM does not noticeably affect Cross-Attention latency (row 5 *vs.* row 6, col 2), and the additional cost of computing the dynamic scaling matrix s is relatively small and well within acceptable bounds (row 5 *vs.* row 6, cols 3–5).

In summary, through the use of efficient modules (MagiAttention) and optimized strategies (*e.g.*, lightweight scaling matrix design), Lumos$\mathcal{X}$ achieves high computational efficiency with minimal additional inference overhead (row 6 *vs.* rows 1 and 4).

Table 13: Inference-stage computational overhead and latency statistics for each module under the same video customization case on an H20 GPU.

| Methods | Self-Attn Latency | Cross-Attn Latency | Latency/step | FLOPs/step | GPU Memory Usage |
|---|---|---|---|---|---|
| None | 0.1440s | 0.0026s | 8.66s | 195.44 T | 21.5G |
| +R2PE | 0.1441s | 0.0025s | 8.66s | 195.44 T | 21.5G |
| None (MagiAttention in Self-Attn) | 0.0935s | 0.0025s | 5.79s | 195.44 T | 21.5G |
| +R2PE (MagiAttention in Self-Attn) | 0.0936s | 0.0026s | 5.79s | 195.44 T | 21.5G |
| +R2PE+CSAM | 0.0966s | 0.0026s | 5.81s | 195.44 T | 21.5G |
| +R2PE+CSAM+MCAM | 0.0965s | 0.0045s | 6.11s | 195.46 T | 22.7G |

## D  More Visualization Results

### D.1  Additional Results of Identity-Consistent Video Generation

The qualitative identity-consistent video generation results are shown in Fig. 14. Under the single-subject setting (Case 1 and Case 2), our method demonstrates significantly superior identity preservation performance compared to ConsisID (Yuan et al., 2024) and Concat-ID (Zhong et al., 2025). Under the multi-subject setting (Case 3 and Case 4), our approach demonstrates superior performance over SkyReels-A2 (Fei et al., 2025) and Phantom (Liu et al., 2025) in both identity preservation and the alignment between multiple subjects and captions, effectively preventing character confusion (Case 3 SkyReels-A2) and positional confusion (Case 3 Phantom).

### D.2  Additional Results of Subject-Consistent Video Generation

The qualitative comparison of subject-consistent video generation is shown in Fig. 15 and Fig. 16. The experimental results demonstrate that our model supports flexible multi-subject foreground-background video customization. Compared to SkyReels-A2 (Fei et al., 2025) and Phantom (Liu et al., 2025), our approach achieves superior subject consistency, accurately matching the human faces and their corresponding attributes, and maintaining the reference identity throughout the generation process. In contrast, SkyReels-A2 struggles when handling multiple customized subjects, often exhibiting character confusion (Cases 2, 4 in Fig. 15 and Case 1 in Fig. 16) and subject disappearance (Case 3 in Fig. 15). Similarly, Phantom also suffers from character confusion (Cases 1,2 in Fig. 16) in comparable scenarios. Furthermore, the videos generated by our Lumos$\mathcal{X}$ are natural and realistic. Unlike Phantom, which suffers from noticeable quality degradation as the number of reference condition images increases, resulting in visual artifacts (Case 1 in Fig. 15 and Cases 2, 3, 4 in Fig. 16) or an unintended cartoon-like style (Cases 2, 3, 4 in Fig. 15).

## E  Limitations and Future Work

Although our Lumos$\mathcal{X}$ effectively addresses the key challenge of multi-subject video personalization by explicitly modeling the dependency of face–attribute within the subject group, it remains constrained by limitations in model size as well as the diversity and scale of training data. Consequently, the performance of Lumos$\mathcal{X}$ has not yet reached its full potential. Looking ahead, we plan to deploy Lumos$\mathcal{X}$ on Wan2.1-14B-T2V model (Wang et al., 2025) and train it on a larger-scale, higher-quality, and more diverse dataset to further enhance its performance and generalization capabilities. Moreover, to push the boundaries of dynamic behavior understanding and better capture complex motion patterns, we also recognize the value of incorporating motion-aware constraints—for instance, augmenting data collection with motion descriptions (*e.g.*, walking, running) and integrating motion cues within the MCAM module to strengthen correlations between visual tokens and motion-aware textual tokens, which would improve alignment for dynamic behaviors and multi-subject interactions (*e.g.*, hugging, handshaking, passing objects).

## F  LLM Usage Statement

We used large language models (LLMs) to assist with the writing and refinement of this manuscript, primarily for improving grammar, clarity, and overall readability. The LLMs were not involved in formulating research questions, designing experiments, developing algorithms, conducting analyses,

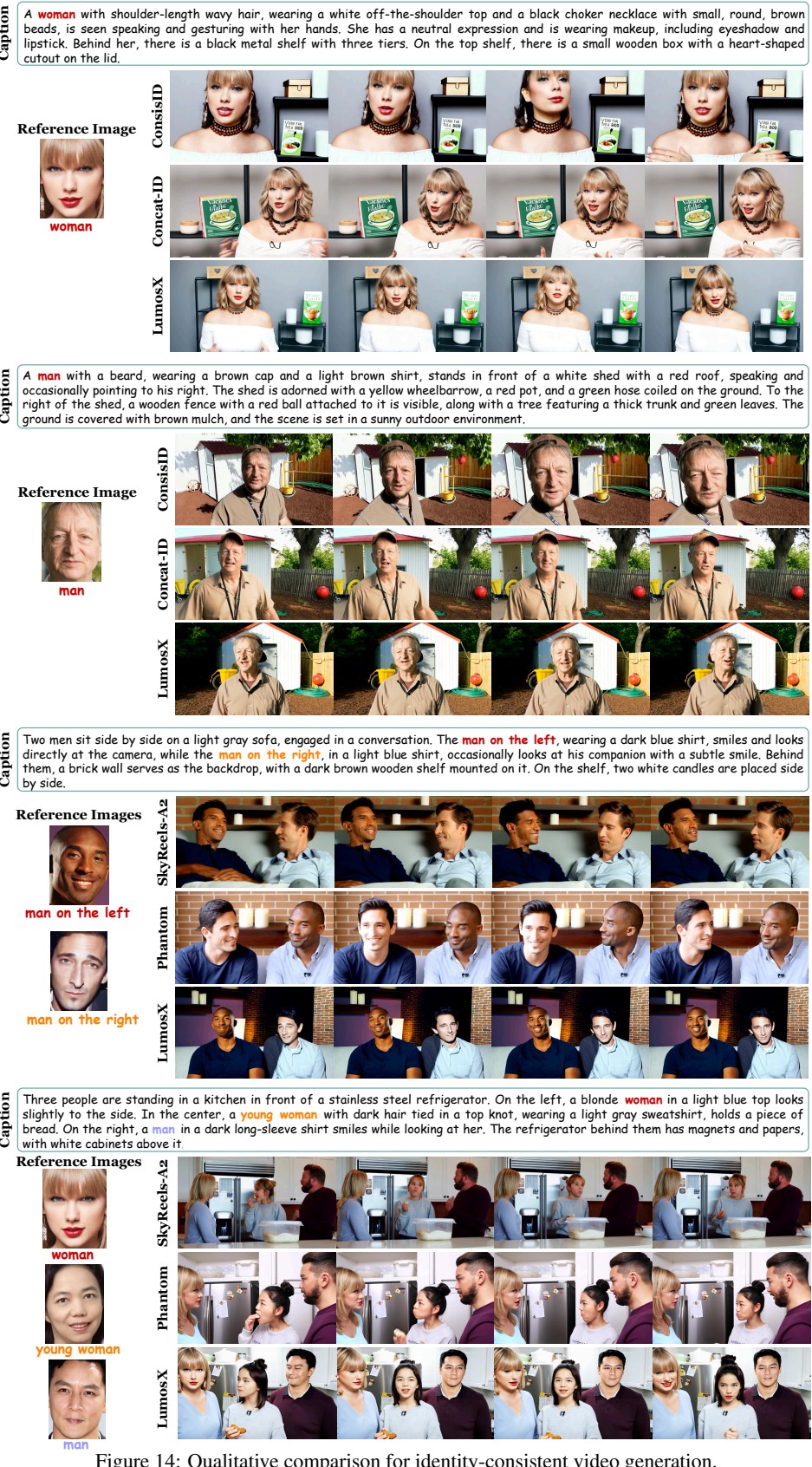

Figure 14: Qualitative comparison for identity-consistent video generation.

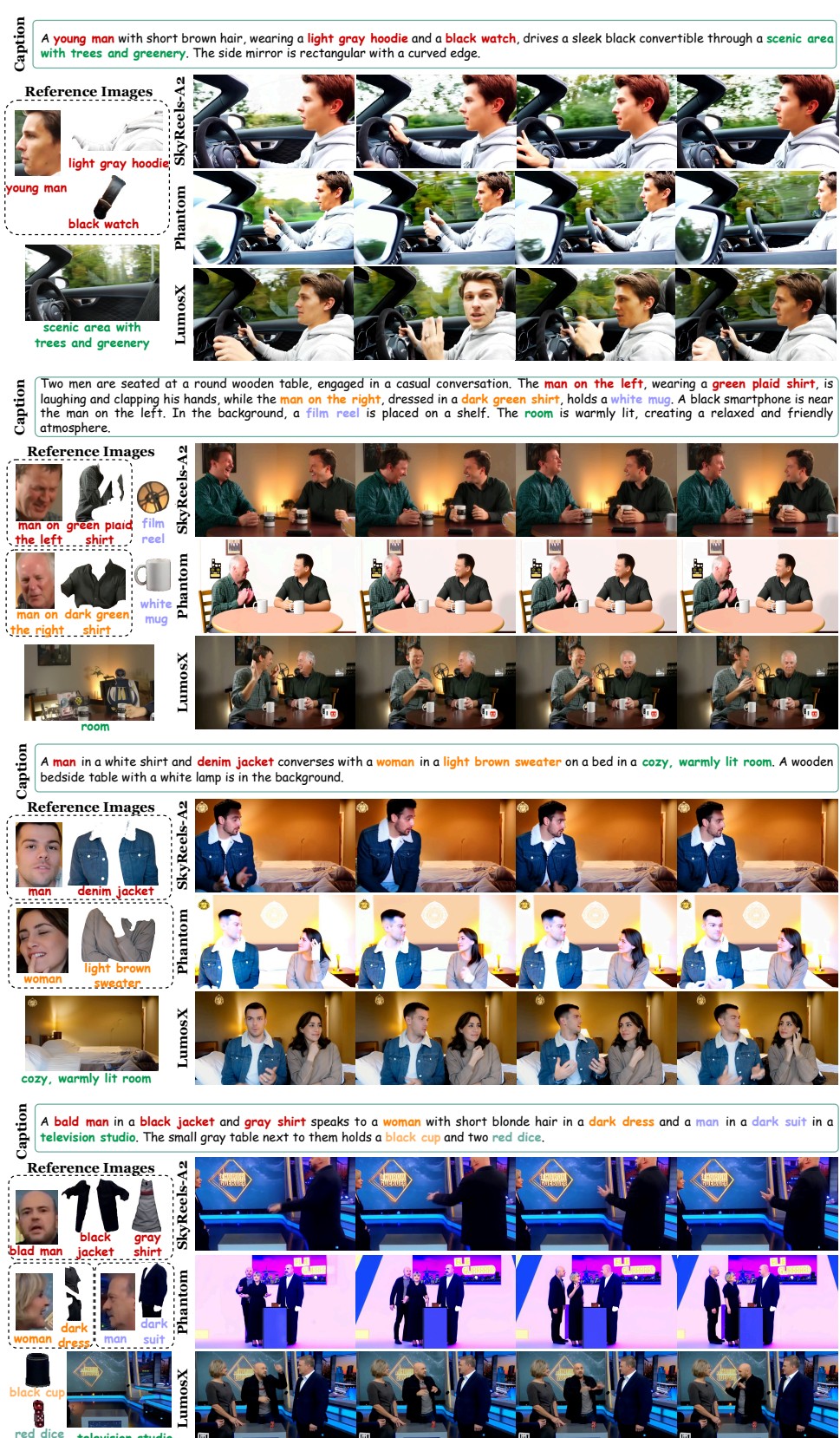

Figure 15: Qualitative comparison for subject-consistent video generation.

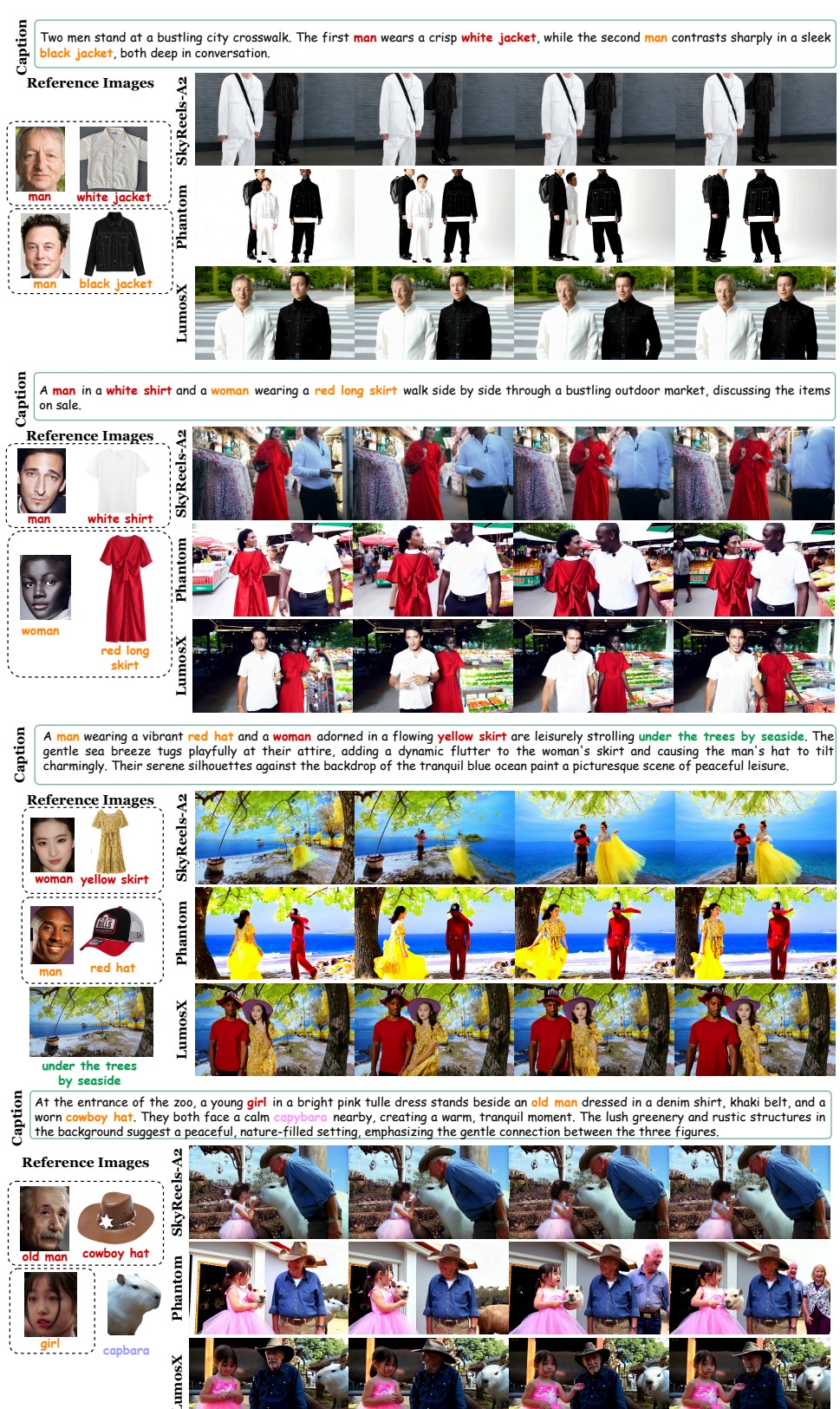

Figure 16: Qualitative comparison for subject-consistent video generation.

or drawing conclusions. All substantive content, including the conceptual framework, methodology, experiments, and findings, is entirely our own and has been independently verified by the authors.

## G  BOARDER IMPACT

This work proposes a personalized multi-subject video generation framework capable of flexibly customizing both the foreground and background of generated videos. It demonstrates strong potential in various application domains such as virtual theatrical production and e-commerce. We recognize the ethical and social implications of generative content, particularly concerning content authenticity, potential misuse, and privacy. As our method explicitly focuses on identity generation and face synthesis, we emphasize the importance of responsible use in accordance with relevant legal and ethical standards. Meanwhile, all training data utilized in our framework were obtained from publicly available datasets (Panda70M) under appropriate usage licenses. We also encourage future research to incorporate safeguards such as digital watermarking, provenance tracking, and content verification tools to support the trustworthy deployment of personalized video generation models.

