# OpenReview forum: "LumosX: Relate Any Identities with Their Attributes for Personalized Video Generation"
_ICLR.cc/2026/Conference — ICLR 2026 Poster_

### Official Review · Reviewer_sFZt · 2025-10-15

**Soundness:** 3
**Presentation:** 3
**Contribution:** 3
**Rating:** 8
**Confidence:** 4

**Summary:**

This paper addresses **personalized multi-subject video generation** where each subject is defined not only by identity (face) but also by **subject-specific attributes** (e.g., clothing, accessories) and an explicit **background/object** context. On the data side, the authors build a complete pipeline that re-captions videos, extracts entities/attributes, and produces **reference image groups** for faces, per-subject attributes, independent objects, and a clean background—then binds these to **prompt tokens** for training. On the modeling side (built on Wan 2.1, 1.3B), they introduce three key components—**R2PE**, **CSAM**, and **MCAM**. Experiments focus on both identity consistency and **multi-subject group consistency**, with ablations indicating each component’s contribution.

**Strengths:**

- **Clear problem focus & delta.** Goes beyond prior “multi-face-only” personalization to **multi-subject** customization with explicit attribute and background/object control, making it applicable to more realistic scenes.
- **Well-engineered data pipeline.** The end-to-end collection & preprocessing pipeline—especially **binding reference images to prompt tokens**—is practical and valuable; if released, it would benefit future work.
- **Methodological novelty.** The proposed **R2PE**, **CSAM**, and **MCAM** are sensible and targeted to the identified failure modes; ablations support their effectiveness.
- **Stronger multi-subject results.** Improvements are most evident in **multi-subject group** consistency compared to recent baselines.
- **Reproducibility mindset.** Implementation details are reasonably complete; the paper frames a clear benchmark setting derived from the same pipeline.

**Weaknesses:**

- **Baseline parity/inputs.** It’s not always clear that baselines ingest the same **attribute/object/background** references rather than face-only cues; this affects fairness in multi-subject comparisons.

**Questions:**

1. **Subject-consistency setting vs. baselines.** Both **SkyReels-A2** and **Phantom** claim multi-subject capability. In your **subject-consistency** experiments, did they receive exactly the same bundle of inputs (per-face, per-attribute, per-object, and background references), or only faces? If not identical, please clarify the input protocol and why the comparison is fair.

---

> ### Author Response · Authors · 2025-11-25
> **Response to Reviewer sFZt**
>
> We sincerely appreciate the reviewer’s thoughtful and insightful feedback. We are also thankful for the positive remarks on our problem focus, data collection pipeline design, experimental results, and reproducibility mindset. Below, we provide detailed responses to each comment and highlight all revisions made to both the main paper and the Appendix (in blue).
>
> **W1&Q1:** **Clarification of Baseline Input Parity in Multi-Subject Comparisons**
>
> **A:** Thank you for raising this important point regarding input fairness. In our Subject-Consistent Video Generation experiments, all baselines receive exactly the same input data as LumosX. This includes:
>
> - Each subject's face image
> - All subjects' corresponding attribute images
> - All object-level reference images
> - The background reference image
> - A shared global text prompt
>
> We strictly follow an identical input protocol across all models to ensure fair comparison.
> In addition, for the Identity-Consistent Video Generation setting, the input protocol is also unified:
>
> - Each subject's face image
> - A shared global text prompt
>
> No attribute, object, or background images are provided in this setting for any method.
> We have included this full description of the input protocols in the revised main paper (Sec. 4.2, lines 406–410) to ensure full transparency of the evaluation setup and to clearly justify the fairness of our comparisons.

---

### Official Review · Reviewer_GYqr · 2025-10-20

**Soundness:** 2
**Presentation:** 4
**Contribution:** 2
**Rating:** 4
**Confidence:** 5

**Summary:**

This paper introduces LumosX, a multi-subject video personalization framework which aims at maintaining the identity and attribute consistency across subjects. To achieve this, LumosX integrates both data-level and model-level innovations: 1) on the data side, the authors uses Qwen2.5-VL to infer subject–attribute relations and generate explicit relational priors; 2) on the modeling side, they introduce relational self-attn & cross-attn, which incorporate relational RoPE and custom attention masks to enhance intra-group cohesion and suppress cross-group interference. Extensive experiments show that LumosX achieves SOTA performance compared to the baseline models like Phantom and SkyReels-A2, particularly in maintaining identity consistency and semantic alignment.

**Strengths:**

- This paper focuses the problem of identity-attribute consistency, which sounds interesting but has been well explored topic in the domain of multi-subject video personalization.
- From the Tables 1~3, LumosX shows strong quantitative results comparing the SOTA models, like Phantom and SkyReels.
- The paper is well written and easy to follow. The figures are well-plotted and informative which can make readers quickly understand the core ideas.

**Weaknesses:**

- Although the paper focuses on maintaining consistent identity-attribute pairing, the author only show one generated example with multiple identity-attribute pairs (the right example in Figure 5). With such one sample, it is unconvincing to claim that LumosX is capable of solving the issue of inconsistent identity-attribute pairing.
- While interesting, it is unclear whether the issue of inconsistent identity-attribute pairing is an actual problem. First, in the only multi-subject example provided in the paper, both the comparing models (Phantom and SkyReels) do not exhibit attribute-swapping issues. Further, most of the existing models also supports full-body personalization which can easily fix such problem without the proposed relational design. If the problem rarely appears or can be easily fixed by the current models, the value of the proposed framework is limited.
- The relational modeling is too simplistic and only considers pairwise identity-attributes grouping. Therefore, the model cannot handle more complex spatial or interaction relations, which are quite common in real-word videos. For example, "the person holding the cup" vs. "the cup near the person" (which requires the spatial modeling) or "person-person or attribute-attribute" interactions.
- Another problem of the proposed relational modeling is that it is determined during the dataset collection phase. Such design assumes the relations are fixed throughout the video and fails to handle more complex scenarios where the scene evolves, such as someone turns around / changes pose / are occluded. The authors could consider dynamic re-binding design which could address the limitation by adjusting the grouping relation during video generation process.
- While the paper introduces multiple masks, such as CSAM and MCAM, to bias the attention, it is unclear how attention actually changes under these biases. The author could visualize the attention maps or compute the attention scores to better interpret / understand the proposed relational self-attn / cross-attn.
- All experiments are conducged on a private benchmark collected by the authors themselves, which makes it hard to reproduce the results. Could the authors also provide the scores on the public benchmarks, such as MSRVTT-personalization [1] and ​​OpenS2V-5M-Eval [2]?

[1] "Multi-subject Open-set Personalization in Video Generation", CVPR 2025

[2] "A Detailed Benchmark and Million-Scale Dataset for Subject-to-Video Generation", NeurIPS 2025 D&B

**Questions:**

- Could the authors provide more examples with multiple identity-attribute pairs? Since this is the main focus of the paper, it is difficult to justify the model performance without such samples.
- Could the authors also provide the examples that the existing models fail on handling identity-attribute consistency?
- Could the authors discuss the extension with more complex relational and dynamic modeling? The current relational modeling is too simplistic to cover the complexity of real-world videos.
- Could the provide more theoretical/empirical analysis on the proposed CSAM and MCAM?
- Could the authors also provide the scores on the public benchmarks?
- [Minor] The "3D VAE Decoder" block in Figure 3 is upside-down.
- [Minor] What is CSMA in Section 4.3 and Appendix C?

---

> ### Author Response · Authors · 2025-11-25
> **Response to Reviewer GYqr (1/3)**
>
> We sincerely thank the reviewer for meticulous and comprehensive critique. We also appreciate the positive remarks regarding our results and presentation. Below, we address each concern in detail and indicate all revisions made to both the revised main paper and the Appendix (in blue), together with an updated demo.
>
> **W1&Q1: Insufficient Evidence for Consistent Face–Attribute Pairing**
>
> **A:**  We thank the reviewer for the helpful suggestion. Due to space limitations in the revised main paper, we were only able to include one representative example. However, additional visual results demonstrating multiple face–attribute pairings are provided in revised Appendix Fig. 15 and Fig. 16, where the superiority of our method in maintaining accurate face–attribute matching can be clearly observed. In the revised version, we also include two additional multi-subject video generation cases in Fig. 5(b). Furthermore, we have prepared a summary comparison video for the supplementary material, which more intuitively showcases the strengths and weaknesses of different methods, including the consistent identity–attribute pairing achieved by our approach.
>
> **W2&Q2: Questioning the Practical Relevance of the Face–Attribute Pairing Problem**
>
> **A:** We appreciate the reviewer’s thoughtful comments. In fact, LumosX was motivated precisely because existing methods—such as SkyReels-A2 and Phantom—frequently exhibit inconsistent face–attribute pairing in multi-subject personalization, which we observed repeatedly during our experiments. This issue is not hypothetical; it occurs in practice and becomes more severe as the number of subjects increases.
> To address the reviewer’s concern about evidence, we provide multiple visual cases in the revised appendix that clearly demonstrate attribute-swapping or mismatched face–attribute associations:
>
> - SkyReels-A2: Revised main paper Fig.5(b) (cases 3 and 4); revised Appendix Fig. 14 (cases 3 and 4), Fig. 15 (cases 2 and 4), Fig. 16 (cases 1 and 2).
> - Phantom: Revised main paper Fig.5(b) (case 3); revised Appendix Fig. 14 (case 3), Fig. 15 (case 2), Fig. 16 (case 2).
>
> Beyond face–attribute mismatches, existing methods consistently suffer from significant video degradation in multi-subject settings. Examples include:
>
> - SkyReels-A2: Revised Appendix Fig. 16 (case 3)
> - Phantom: Revised main paper Fig. 5(b) (cases 2 and 4), revised Appendix Fig. 13 (cases 2–4), and Fig. 14 (case 3)
>
> Additionally, both baselines often produce poor facial similarity, which violates the core requirement of multi-subject personalization:
>
> - SkyReels-A2: Revised main paper Fig. 5(b) (cases 2 and 4), revised Appendix Fig. 14 (cases 3 and 4), Fig. 15 (cases 2-4), Fig. 16 (cases 1-4).
> - Phantom: Revised main paper Fig. 5(b) (cases 2 and 4), revised Appendix Fig. 14 (case 4), Fig. 15 (case 4), Fig. 16 (cases 1–4).
>
> Taken together, these results demonstrate that inconsistent identity–attribute pairing and degraded identity preservation are real, recurring problems in current models. Therefore, the relational design in LumosX directly addresses a practically significant and empirically observed challenge.
>
> **W3&Q3: Limitations of Pairwise Relational Modeling**
>
> **A:** We appreciate the reviewer's insightful comment. We agree that our current relational modeling—focused on face–attribute pairings—does not explicitly capture more complex spatial or interaction relations. As discussed in revised Appendix Sec. E, extending LumosX with motion-aware constraints is a natural and promising direction toward addressing these richer relational structures.
> In particular, incorporating motion-aware cues into our relational attention framework would allow the model to better handle cases such as *"the person holding the cup"* or multi- subject interactions. Based on our current architecture, such enhancements could be integrated as follows:
>
> - Data Collection: Motion descriptions (e.g., *walking*, *running*) can be appended to each subject group's textual tags to provide explicit behavioral context.
> - MCAM Module: Motion cues can be added to the textual subject groups to expand the Strong Correlation regions (Eq. 3, revised main paper), enabling stronger attention links between visual tokens and motion-aware textual tokens. This would improve alignment for spatially dependent or dynamic actions.
> - Multi-Subject Dynamics: For interactions such as *hugging*, *handshaking*, or *passing objects*, MCAM can be extended with an additional correlation category designed to explicitly model inter-subject motion dynamics.
>
> We thank the reviewer again for highlighting this valuable direction. We view motion-aware relational modeling as a highly promising extension of LumosX and plan to explore it in future work.

---

> ### Author Response · Authors · 2025-11-25
> **Response to Reviewer GYqr (2/3)**
>
> **W4: Lack of Dynamic Relational Re-Binding in Evolving Scenes**
>
> **A:** Thank you for the insightful suggestion. We agree that dynamic relational re-binding is a promising direction, especially for scenarios where subject–attribute relations evolve over time (e.g., turning, pose changes, or occlusions). However, LumosX is built on a diffusion-based framework, which processes all frames jointly in a single denoising trajectory. This global generation paradigm makes it difficult to incorporate dynamic re-binding, as the model does not generate frames sequentially. That said, we acknowledge the value of dynamic re-binding, and we note that such a mechanism is more naturally supported by autoregressive video generation models, which generate frames step-by-step and can therefore adjust grouping relations dynamically as the video evolves. We consider this an exciting future direction and plan to explore it in subsequent work. We aim to investigate autoregressive video generation architectures with a persistent memory mechanism capable of storing subject representations, tracking their temporary disappearance and reappearance, and retrieving the correct identity–attribute bindings as the video evolves. Such a memory-driven, dynamically re-bindable design offers a promising path toward handling more complex and continuously changing multi-subject scenarios.
>
> **W5&Q4: Lack of Visualization of Attention Changes Under CSAM/MCAM**
>
> **A:** Thank you for the valuable suggestion. In the revised Appendix Sec. C.7, we have added a dedicated Visualization of Attention Changes Under CSAM/MCAM to illustrate how these masks influence the relational self-attention and cross-attention mechanisms. The results in this figure can be compared with those in Fig. 4 of the revised main paper. From this comparison, we observe that CSAM (Fig. 10(a) vs. Fig. 10(b), unpdated Appendix)  enables the denoising branch to independently aggregate conditional signals and effectively bind face–attribute dependencies within the conditional branch. Additionally, MCAM (Fig. 10(c) vs. Fig. 10(d), unpdated Appendix) explicitly enhances both intra-group and inter-group correlations among subject groups and further strengthens the semantic-level representation of the visual condition tokens. These visualizations help clarify how our relational masks modulate attention behavior and support the design choices in our model.
>
> **W6&Q5: Missing Evaluation on Public Personalization Benchmark**
>
> **A:** Thank you for the suggestion. To further evaluate our method on public benchmarks, we conducted a quantitative comparison between our LumosX and existing approaches, including SkyReels-A2 and Phantom, on the MSRVTT-personalization benchmark. The MSRVTT-personalization benchmark provides two evaluation modes: subject-mode and face-mode. Face-mode focuses solely on face-similarity metrics, whereas we are interested in assessing the overall subject performance (face + attributes). Therefore, we conduct our evaluation under the subject-mode setting. Our evaluation is conducted using one reference image for the subject and one for the background, and the results are summarized in the table below.
>
> | **Method**  | **#Base Model** | **Text-S** | **Vid-S** | **Subj-S** | **Dync-D** |
> | ----------- | :-------------: | :--------: | :-------: | :--------: | :--------: |
> | SkyReels-A2 | Wan2.1-14B-T2V  |   0.253    | **0.781** | **0.554**  |   *0.783*    |
> | Phantom     | Wan2.1-1.3B-T2V | **0.270**  |   0.696   |   0.534    |   0.461    |
> | **LumosX**  | Wan2.1-1.3B-T2V |   *0.258*    |   *0.707*   |   *0.549*    | **0.786**  |
>
> It is worth noting that MSRVTT-personalization is a single-subject benchmark, and its definition of 'subject' differs from ours: a subject may refer to either a human category (e.g., man, woman) or an object category (e.g., car, horse, clothes, hat). Moreover, for human subjects under the benchmark's subject-mode setting, the subject is provided as a holistic entity without decoupled face and attribute references, and therefore does not involve the face–attribute matching problem that our method is designed to address. As a result, our method does not benefit from its explicit face–attribute relational modeling in this setting. In addition, both LumosX and Phantom are built on the Wan2.1-1.3B-T2V base model, whereas SkyReels-A2 adopts the much stronger Wan2.1-14B-T2V model and therefore further benefits from a significantly more capable backbone. Despite these disadvantages, LumosX still achieves the second-best overall performance, demonstrating the robustness and generalization ability of our approach even outside its primary multi-subject setting. We have included this discussion in the revised Appendix Sec. C.6 of the revised version.

---

> ### Author Response · Authors · 2025-12-02
> **Response to Reviewer GYqr (3/3)**
>
> **Q6: 3D VAE Decoder Block Orientation**
>
> **A:** Thank you for pointing this out. We have corrected the orientation of the "3D VAE Decoder" block in Fig. 3 in the revised main paper.
>
> **Q7: What is CSMA in Sec. 4.3 of the main paper and Appendix Sec. C?**
>
> **A:** Thank you for catching this mistake. The term "CSMA" in Sec. 4.3 of the main paper and  Appendix Sec. C is a typographical error. The correct name should be CSAM (Causal Self-Attention Mask). We have fixed this in the revised version to avoid confusion.

---

### Official Review · Reviewer_Mi1n · 2025-10-29

**Soundness:** 2
**Presentation:** 3
**Contribution:** 2
**Rating:** 6
**Confidence:** 1

**Summary:**

This paper proposes LumosX, a framework for personalized multi-subject video generation that addresses the critical challenge of face–attribute misalignment in existing text-to-video (T2V) models. The core innovation lies in explicitly modeling face-attribute dependencies at both the data and model levels, enabling fine-grained, identity-consistent, and semantically aligned video synthesis.

**Strengths:**

1. Combines MLLM-driven data annotation with relational attention to solve a misalignment problem—an innovative fusion of NLP (entity extraction) and computer vision (positional embedding/attention masking) techniques.
2. The problem statement (face-attribute misalignment) is clearly articulated with examples (e.g., "A man on the left... and a man on the right..." causing confusion).
3. Enables flexible multi-subject customization (foreground/background control) that prior models lack.

**Weaknesses:**

1. The training dataset only includes 1–3 subjects, the paper acknowledges instability for 10+ subjects due to RoPE extrapolation. No preliminary results or mitigation detailsare provided beyond a future work note.
2. All evaluations rely on automated metrics. Human judgment of face-attribute alignment, video naturalness, and prompt adherence would strengthen claims (e.g., do viewers perceive fewer misassignments in LumosX-generated videos?).

**Questions:**

1. Do you plan to conduct a human study to assess perceived face-attribute alignment, video naturalness, or prompt adherence? If not, why do you believe automated metrics alone are sufficient to validate LumosX’s practical utility?
2. Can you provide quantitative metrics to support FLUX’s "superior realism" over other inpainting models? How do background quality variations affect downstream video generation?

---

> ### Author Response · Authors · 2025-11-25
> **Response to Reviewer Mi1n (1/2)**
>
> We sincerely thank the reviewer for meticulous and insightful remarks. We are grateful for the encouraging feedback regarding our problem setup, methodological framework, and experimental results. Below, we address each concern in detail and indicate all updates made to the revised main paper and the Appendix (in blue).
>
> **W1:** **Insufficient Discussion of High-Subject-Count Scenarios**
>
> **A:** We appreciate the reviewer's insightful comment. We conducted an exploratory attempt to extend generation to 10+ subjects and provide corresponding visualizations. However, the results were highly unstable and far from satisfactory. We believe this limitation is not only due to RoPE extrapolation but also closely related to the underlying base model. For example, Wan2.1-1.3B-T2V, which we rely on, also struggles significantly when generating videos containing more than ten distinct human subjects. As further evidence, the revised Appendix Sec. C.4 (Fig. 8) provides visualizations of Wan2.1-1.3B-T2V under a 10-subject setting, where the facial quality deteriorates substantially, and the model fails to strictly follow the prompt, generating only nine people instead of ten. This reinforces our point that the base model's capability is the primary bottleneck when extending LumosX to scenarios involving substantially more subjects. These results indicate that current video-generation models like Wan-2.1 lack the capacity to learn stable representations for such large numbers of subjects, largely because high-quality 10+ subject videos are extremely rare. To the best of our knowledge, no existing model can robustly handle scenarios with this many subjects at present. We will highlight this limitation more clearly and discuss potential mitigation strategies in future work.
>
> **W2&Q1:** **Lack of Human Study for Evaluation**
>
> **A:** Thank you for the valuable suggestion. To complement automated metrics, we conducted a user study covering a random sample of 24 video cases, including 6 single-subject, 12 two-subject, and 6 three-subject customization scenarios. The comparison includes LumosX, SkyReels-A2, and Phantom, with a total of 30 participants. Each participant evaluated videos along four dimensions:
>
> - Face–Attribute Alignment: whether each face is correctly matched with its corresponding attributes (e.g., clothing, accessories, or hairstyle).
> - Face Similarity: how closely each generated face resembles the provided visual reference.
> - Video Naturalness: the overall visual quality and coherence of the generated video.
> - Prompt Adherence: whether the generated video follows the instructions specified in the text prompt.
>
> For each case, participants ranked the three methods, and scores were assigned as follows: 1 point for first place, 0.5 points for second place, and 0 points for third place. Final scores for each dimension were computed as the weighted average across all participants. The results are shown table below.
>
> | Methods     | Face–Attribute Alignment | Face Similarity | Video Naturalness | Prompt Adherence | **Average** |
> | ----------- | :----------------------: | :-------------: | :---------------: | :--------------: | :---------: |
> | Skyreels-A2 |          0.281           |      0.276      |       0.317       |      0.330       |    0.301    |
> | Phantom     |          0.516           |      0.463      |       0.456       |      0.479       |    0.478    |
> | **LumosX**  |        **0.703**         |    **0.761**    |     **0.727**     |    **0.691**     |  **0.720**  |
>
> Overall, our method achieves superior performance across all four evaluation dimensions.
> We have included these human-study results in the revised Appendix (Sec. C.12) to provide a more comprehensive evaluation of alignment and video quality.

---

> ### Author Response · Authors · 2025-11-25
> **Response to Reviewer Mi1n (2/2)**
>
> **Q2: Insufficient Quantitative Evaluation of FLUX Realism and Its Influence on Video Generation**
>
> **A:** Thank you for raising this important point. To quantitatively assess the realism of FLUX compared with other inpainting models, we conducted a large-scale evaluation on 2,130 test cases, applying both FLUX and Stable-Diffusion-2 for inpainting. We then computed FID scores against the COCO 2017 Val set to measure distributional realism. In addition, we randomly sampled 100 cases and asked GPT-4o to independently judge which inpainting result appeared more realistic, with details provided in the revised Appendix Sec.C. 11. The results of both evaluations are summarized in the table below.
>
> | Methods                       |    FID    | AI Judgement |
> | ----------------------------- | :-------: | :----------: |
> | Stable-Diffusion-2-Inpainting |   96.32   |     36%      |
> | FLUX-Inpainting               | **92.83** |   **64%**    |
>
> The results show that FLUX achieves better performance under both evaluation metrics. To further understand how background quality affects downstream video generation, we performed a qualitative comparison using the two inpainting outputs as inputs to video generation. As shown in the updated Appendix Fig. 12, artifacts introduced during background inpainting are clearly propagated into the generated video, degrading overall video quality. This confirms that inpainting realism plays a crucial role in high-quality video generation and motivates our choice of FLUX as the inpainting module.

---

### Official Review · Reviewer_mVQz · 2025-10-31

**Soundness:** 3
**Presentation:** 3
**Contribution:** 3
**Rating:** 6
**Confidence:** 4

**Summary:**

This paper presents a novel framework for personalized video generation with controllable facial attributes. The authors introduce a dedicated dataset curation pipeline tailored to this task and propose model designs to capture subject-attribute dependencies. Experimental results demonstrate satisfactory performance.

**Strengths:**

- The paper addresses an important and challenging problem in video generation: enabling both personalization and semantic control, areas where previous works have struggled.
- The paper introduces the dataset for this task and the designs on model side.

**Weaknesses:**

- The focus on generating videos based solely on facial attributes may limit the method’s generalizability to broader contexts.
- It is unclear how the approach performs when handling videos with multiple subjects (three or more), as the datasets support up to three subjects.
- The implementation details lack specifics on computational requirements such as GPU count and training duration.
- The definition of a "subject" during data curation is ambiguous. Figures 3 and 4 suggest that a subject may consist of a single face and its corresponding attribute, but the criteria for data curation and the types of attributes used require further clarification. While some details are mentioned in the appendix, a clear and formal definition should be included in the main text.
- The maximum number of attributes that can be assigned to one subject is not specified.
- It remains unclear whether the method supports using textual attributes in the absence of images. Although text prompts may control generation, the paper would benefit from further discussion and experiments on text-based attribute control in the context of multi-subject video personalization.

**Questions:**

- In the current data curation pipeline, human subjects are extracted from frames at the start, middle, and end of the video, and all input subjects appear in the full dataset. How does the method handle scenarios where a particular subject is present only partway through a video (e.g., featured in the first half but absent in the second)? While autoregressive generation may naively address this by using a sliding window of subjects, clarification on handling such cases would be helpful.
- Since attributes are provided as images, can existing multi-subject image personalization models be used to generate the initial image, followed by video generation? A comparison with such an approach would strengthen the experimental evaluation.

---

> ### Author Response · Authors · 2025-11-25
> **Response to Reviewer mVQz (1/3)**
>
> We sincerely thank the reviewer for the thoughtful and constructive suggestions. We appreciate the affirmative feedback on our motivation and model design. Below, we provide point-by-point responses to the raised concerns and highlight all revisions made to both the main paper and the Appendix (in blue), along with an updated demo.
>
> **W1: Generalization Concerns from the Face-Attribute Focus**
>
> **A:** We sincerely thank you for the thoughtful feedback. Although LumosX focuses on modeling face-attribute relational bindings for subjects, it also supports video customization of foreground objects and background scenes beyond the foreground subjects (face–attribute pairs). This capability is demonstrated across multiple parts of the paper: in data collection (Fig. 2, revised main paper), model design (Figs. 3 and 4, revised main paper), and qualitative results (Fig. 5(b), revised main paper; Figs. 15 and 16, revised Appendix). These examples collectively demonstrate that our framework supports diverse foreground and background configurations, rather than being limited to face–attribute pairs alone.
>
> **W2: Concerns of handling 3+ subjects**
>
> **A:**  LumosX is capable of customizing videos with more than three subjects without retraining. Although the training dataset includes videos with up to three subjects, the model architecture supports generation with a greater number of subjects during inference. Naturally, as the number of subjects increases, the resulting scene complexity may affect model performance, particularly when dealing with cases that go beyond the number of subjects encountered during training. A detailed discussion is provided in the revised Appendix Sec. C.4, where we also include quantitative comparisons for customized videos with three and four subjects (revised Appendix Tab. 6) as well as qualitative visualizations for three-subject cases (Fig. 14 and Fig. 15, last row "case 4").  Additionally, we provide a three-subject example in the revised main paper (Fig. 5(b), lower right), and further three-subject video comparisons can be seen in our updated demo video (cases 6, 12, and 13).
>
> **W3: Lack of Details on Computational Requirements**
>
> A:  Thank you for pointing this out. We have clarified these computational details in the updated Sec. 4.1 (Implementation Details) of the revised main paper. Our full training process required approximately 883 GPU-days on H20 GPUs.
>
> **W4: Lack of a Clear Subject Definition in the Main Paper**
>
> **A:** We thank the reviewer for the helpful suggestion. In our work, a subject is defined as a single human face paired with its corresponding attributes. The face is expected to present clear facial features without significant occlusion, and the associated attributes can include clothing (top or bottom), accessories (e.g., glasses, earrings, or necklaces), or hairstyle. We have incorporated this formal definition into Sec. 3.2 of the revised main paper for greater clarity.
>
> **W5: Undefined Attribute Capacity for a Single Subject**
>
> **A:** We thank the reviewer for the helpful suggestion. During training, each subject’s face is associated with up to three attributes. Therefore, we recommend providing no more than three attributes per subject group during inference to remain consistent with the training setup. We have also added a clear description of the attribute capacity to Sec. 4.1 (Implementation Details) of the revised main paper.

---

> ### Author Response · Authors · 2025-11-25
> **Response to Reviewer mVQz (2/3)**
>
> **W6: Lack of Clarity on Text-Based Attribute Control Without Images**
>
> **A:**  We appreciate the reviewer’s insightful comment. Our method does support attribute control using text alone, and we have already included this setting in both the revised main paper and the Appendix under Identity-Consistent Video Generation (Sec. 4.1,  revised main paper). In this setting, we provide only the facial visual image, while all attribute specifications are controlled solely through text prompts. Quantitative evaluations of identity preservation are reported in Tab. 1 and Tab. 2 (revised main paper), and qualitative comparisons are shown in Fig. 5(a) (revised main paper) and Fig. 14 (revised Appendix). In addition, we perform subject-region quantitative evaluations under this setting—using the same metrics as in Subject-Consistent Video Generation—to compare text-based attribute control across different methods. The results are presented in the table below.
>
> | Methods     | Attribute Control | CLIP-T | CLIP-I | DINO-I |
> | ----------- | :---------------: | :----: | :----: | :----: |
> | SkyReels-A2 |     text-only     | 0.192  | 0.643  | 0.192  |
> | SkyReels-A2 |    text&visual    | 0.178  | 0.606  | 0.192  |
> | Phantom     |     text-only     | 0.207  | 0.687  | 0.209  |
> | Phantom |    text&visual    | 0.185  | 0.647  | 0.216  |
> | LumosX      |     text-only     | 0.205  | 0.684  | 0.210  |
> | LumosX |    text&visual    | 0.193  | 0.681  | 0.265  |
>
> Overall, the CLIP-T scores are higher under text-only control, which is expected since the attributes are directly specified via text prompts, naturally yielding higher text–video similarity. CLIP-I trends largely mirror those of CLIP-T because both measure semantic similarity. Notably, for LumosX (rows 6-7, col. 4), the performance gap between text-only control and text&visual control is very small, whereas for the other methods (rows 2-5, col. 4), performance under text&visual control drops noticeably compared to text-only control.  This suggests that those methods become more prone to attribute confusion once visual conditions are introduced, likely due to the absence of explicit face–attribute relational modeling—whereas LumosX avoids this issue. For DINO-I, text-only control is clearly weaker than text&visual control in LumosX. At the same time, the substantial improvement under text&visual control indicates that our face–attribute relational constraints are accurately matched and effectively utilized. In comparison, Phantom and SkyReels-A2 do not show notable DINO-I improvements when visual attribute conditions are added, suggesting that their face–attribute alignment is less reliable. Meanwhile, under the text-based attribute control setting, LumosX performs on par with Phantom and clearly outperforms SkyReels-A2, demonstrating its strong generalization ability even without attribute images. We have added this discussion to Sec. C.9 of the revised Appendix.
>
>
> **Q1**: **Concerns Regarding the Handling of Subjects Appearing Only in Part of the Video**
>
> **A:** Thank you for raising this important question. Since LumosX is diffusion-based, handling scenarios where a subject appears only partway through a video (e.g., present in the first half but absent in the second) can currently be achieved only through prompt design, which controls a subject’s entrance or departure from the scene. We experimented with such prompt-based control on both the baseline Wan2.1-T2V-1.3B and our LumosX, but the results were not ideal, with relatively low success rates. We believe this limitation primarily stems from two factors:
>
> - The underlying Wan2.1-T2V-1.3B base model lacks strong fine-grained temporal control, making it difficult to reliably enforce subject appearance or disappearance based solely on textual prompts.
> - Our training data and pipeline assume that each subject remains visible throughout the entire video, which means the model has not been exposed to partial-presence cases and therefore cannot generalize well to such scenarios.
>
> We appreciate the reviewer's suggestion. In future work, we plan to incorporate such more complex cases by collecting data that explicitly includes subject entrances/exits, and by exploring autoregressive video generation frameworks, which may naturally provide better temporal and entity-level control for multi-subject personalization.

---

> ### Author Response · Authors · 2025-11-25
> **Response to Reviewer mVQz (3/3)**
>
> **Q2: Comparison with Image-Personalization–Based Initialization**
>
> **A:** Thank you for the valuable suggestion. To evaluate whether multi-subject image personalization models can serve as an initial image generator prior to video synthesis, we conducted an additional experiment using UNO [1] to produce a multi-subject customized image, which was then fed into Wan2.1-14B-I2V [2] for video generation. We quantitatively compared this pipeline (UNO + Wan-I2V) against our method on our benchmark. The results are shown in the table below:
>
> | Methods               |  Dynamic  | ViCLIP-T  | ViCLIP-V  |  CLIP-T   |  CLIP-I   |  DINO-I   |  FaceSim  |  CurSim   |
> | -------------------- | :-------: | :-------: | :-------: | :-------: | :-------: | :-------: | :-------: | :-------: |
> | UNO + Wan2.1-I2V-14B |   0.547   | **0.261** |   0.880   | **0.201** |   0.650   |   0.197   |   0.237   |   0.244   |
> | **LumosX**           | **0.782** |   0.259   | **0.945** |   0.193   | **0.681** | **0.265** | **0.460** | **0.493** |
>
> Overall, our method outperforms UNO + Wan2.1-I2V-14B, particularly on ViCLIP-V, DINO-I, FaceSim and CurSim, which reflects facial identity consistency. This improvement is expected, as UNO is not specifically designed for face-aware attribute binding, whereas our approach explicitly models face–attribute relational constraints, leading to significantly better identity preservation in video personalization. Our ViCLIP-T and CLIP-T scores are slightly weaker than those of UNO + Wan2.1-I2V-14B, which may be attributed to the latter's use of a substantially larger base model (Wan-14B). We have also added this result and discussion to Sec. C.10 of the revised Appendix.
>
> [1]  Less-to-More Generalization: Unlocking More Controllability by In-Context Generation, ICCV 2025
>
> [2]  Wan: Open and Advanced Large-Scale Video Generative Models, Arxiv 2025

---

### Author Response · Authors · 2025-11-25
**Response Summary**

We sincerely thank all reviewers for their thoughtful feedback and constructive suggestions. In response to the provided comments, we have substantially revised the main paper, expanded the appendix, and updated the demo video to address the raised concerns. The major updates are summarized below:

### 1. Main Paper Revisions

- **Clarified model capabilities and implementation details**, including subject definition, attribute capacity, and computational requirements (Sec. 3.2 and Sec. 4.1).
- **Provided full input protocol descriptions** for both Identity-Consistent and Subject-Consistent settings (Sec. 4.2).
- **Added additional multi-subject customization visualizations** in Fig. 5(b).

###  2. Appendix Updates

- **Public Benchmark Evaluation (Sec. C.6):** Added MSRVTT-personalization comparison results.
- **Attention Visualization (Sec. C.7):** Added CSAM/MCAM attention similarity map visualizations to clearly illustrate their effectiveness in enhancing relational attention.
- **Text-Based Attribute Control (Sec. C.9):** Added new subject-region quantitative evaluations under the text-based attribute control setting, highlighting the robustness of LumosX compared with the attribute confusion observed in baseline methods.
- **Image-Personalization Comparison (Sec. C.10):** Added a quantitative comparison with image-personalization–based initialization, showing that LumosX achieves stronger overall performance than UNO + Wan2.1-I2V-14B.
- **FLUX Inpainting Evaluation (Sec. C.11):** Added FID and GPT-4o assessment results comparing FLUX and Stable-Diffusion-2, along with qualitative evidence showing that artifacts introduced by Stable-Diffusion-2 inpainting propagate into video generation, highlighting the importance of choosing a reliable inpainting model for data collection.
- **Human Study (Sec. C.12):** Added user study where LumosX achieves the highest average score across all four evaluation dimensions.

### 3. Demo Video Updates

- Added **a summary comparison video** that includes several representative cases from both the Identity-Consistent and Subject-Consistent Video Generation settings.

---

### Author Response · Authors · 2025-12-02
**Rebuttal Summary for AC**

We sincerely thank the Area Chair for overseeing our submission and greatly appreciate all reviewers for their constructive and detailed feedback. We are especially encouraged by the recognition of our `well-motivated problem formulation` (mVQz, Mi1n, sFZt), `well-engineered data pipeline`(mVQz, sFZt),`innovative model design` (mVQz, Mi1n, sFZt), `strong empirical results` (Mi1n, GYqr, sFZt), `clear writing` (GYqr), and `reproducibility mindset` (sFZt).

In this work, we made the key contributions:

- **Data Side:** Built an open-set multi-subject data pipeline with explicit face-attribute dependencies, enabling finer-grained relational priors and more reliable benchmarks.
- **Model Side:**  Introduced relational self- and cross-attention, equipped with relational positional encodings and structured masks, to encode face–attribute bindings and improve consistency in multi-subject video customization.
- **Overall Performance:** LumosX achieved SOTA results in fine-grained, identity-consistent multi-subject video generation, outperforming open-source baselines such as Phantom and SkyReels-A2.

To assist the Area Chair, we summarize below our responses to the major concerns and questions raised by the reviewers.
***
### 1. Reviewer mVQz (Rating: 6)

**Main Concerns**: model definitions and implementation details; handling 3+ subjects; text-based attribute control without images; image-personalization-based initialization

**Our Response:**

- **Definitions and Details**: Added a formal definition of the subject (Sec. 3.2) and clarified the attribute capacity and computational requirements (Sec. 4.1).
- **Handling 3+ Subjects:** Provided qualitative and quantitative results for 3–4 subjects (Appendix Tab. 6; Figs. 14–15) and added new 3-subject visualizations in Fig. 5(b) along with an updated demo video.
- **Text-Based Attribute Control**: Added quantitative analyses under text-only and text&visual settings, showing LumosX avoids attribute confusion observed in baselines when visual attributes are introduced (Appendix Sec. C.9).
- **Image-Personalization Comparison:** Added quantitative comparisons showing LumosX outperforms image-personalization–initialized pipelines (Appendix Sec. C.10).

### 2. Reviewer Mi1n (Rating: 6)

**Main Concerns**: high-subject-count scenarios; human study for evaluation; evaluation of FLUX inpainting

**Our Response:**

- **High-Subject-Count Scenarios:** Added a discussion in Appendix Sec. C.4 showing that the limitations in customizing videos with more than 10 subjects are closely tied to the capability of the underlying T2V base model.
- **Human Study:** Added a user study involving 1-, 2-, and 3-subject settings. LumosX achieves the highest average perceptual score across face-attribute alignment, face similarity, video naturalness, and prompt adherence (Appendix Sec. C.12).
- **FLUX Inpainting Evaluation:** Added FID evaluations (2,130 samples), GPT-4o assessments, and qualitative studies demonstrating that weaker inpainting models inject artifacts that propagate into generated videos, supporting our choice of FLUX (Appendix Sec. C.11).

### 3. Reviewer GYqr (Rating: 4)

**Main Concerns**: evidence supporting the importance of face-attribute matching; visualization of attention changes; evaluation on public personalization benchmark; extension of LumosX

**Our Response:**

- **Evidence Supporting the Importance of Face-Attribute Matching:** Highlighted multiple visual cases—both in the main paper and Appendix—showing attribute swapping and mismatched face-attribute bindings in existing methods lacking explicit relational modeling.
- **Visualization of Attention Changes Under CSAM/MCAM:** Added attention similarity map visualizations showing how CSAM/MCAM reshape relational attention and strengthen intra-/inter-group correlations, enabling explicit face–attribute matching (Appendix Sec. C.7).
- **Evaluation on Public Personalization Benchmark:** Added quantitative results on the MSRVTT-personalization benchmark, clarifying its characteristics. LumosX shows competitive performance (Appendix Sec. C.6).
- **Extension of LumosX:** Recognized the value of extending LumosX beyond pairwise face–attribute modeling and proposed motion-aware extensions for richer relational dynamics. Noted that dynamic re-binding is challenging for diffusion models and highlighted autoregressive, memory-based approaches as a promising future extension.

### 4. Reviewer sFZt (Rating: 8)

**Main Concerns**:  unified input protocols

**Our Response:**

- **Unified Input Protocols:** Provided unified input protocols for both evaluation settings to ensure fair comparisons (Sec. 4.2).
***
We hope this summary helps address the major concerns. Full point-by-point responses are provided in the individual threads. We have revised the main paper, expanded the Appendix, and updated the demo video accordingly.

Best regards,

Authors of LumosX

---

### Meta-Review · Area_Chair_t76D · 2025-12-29

**Summary:**

This paper introduces LumosX, a framework designed to improve multi-subject video generation by explicitly modeling face-attribute alignment using multimodal LLMs for data curation and relational attention mechanisms. The reviewers generally recognized its strengths, including the clear problem formulation and the well-engineered data pipeline. However, they also raised concerns regarding the comprehensiveness of the evaluation, specifically the lack of human user study and comparison on public benchmarks. Reviewer GYqr questioned whether the face-attribute swapping issue is actually important in baselines and that the relational modeling may be too simplistic for complex spatial interactions. Additionally, authors are suggested to provide results on scenarios with more than 3 subjects and the performance on identity preservation should be further improved.

**Reviewer Concerns:**

The authors made efforts to address the major empirical concerns by conducting a human user study, evaluating on the public MSRVTT benchmark, and providing clear visual evidence that baselines indeed suffer from attribute swapping. They also clarified the "subject" definition, and demonstrated performance on 3-4 subjects. The remaining limitation includes the model's inability to handle complex dynamic re-binding or more than 10 subjects due to the constraints of the underlying diffusion base model. However, this is an acceptable scope limitation for the current work.

**Reviewer Scores:**

Reviewer sFZt (Score 8) would likely maintain the high score, as the concern regarding input parity is well explained by authors. Reviewers mVQz (Score 6) and Mi1n (Score 6) would likely maintain their scores or increase their scores to a 7, given the addition of the requested human study, implementation details, and comparisons with image-personalization baselines. Reviewer GYqr (Score 4) would likely move to a 5, as the authors provided the requested results on public benchmark and attention map visualization, and explained the problem's significance through experimental results.

---

### Decision · Program_Chairs · 2026-01-26

Accept (Poster)